# Understand What LLM Needs:
# Dual Preference Alignment for Retrieval-Augmented Generation

## Abstract

Retrieval-augmented generation (RAG) has effectively mitigated the hallucination problem of large language models (LLMs). However, the difficulty of aligning the retriever with the LLMs' diverse knowledge preferences inevitably poses a challenge in developing a reliable RAG system. To address this issue, we propose DPA-RAG, a universal framework designed to align diverse knowledge preferences within RAG systems. Specifically, we initially introduce a preference knowledge construction pipeline and incorporate five novel query augmentation strategies to alleviate preference data scarcity. Based on preference data, DPA-RAG accomplishes both external and internal preference alignment: 1) It jointly integrates pairwise, pointwise, and contrastive preference alignment abilities into the reranker, achieving external preference alignment among RAG components. 2) It further introduces a pre-aligned stage before vanilla Supervised Fine-tuning (SFT), enabling LLMs to implicitly capture knowledge aligned with their reasoning preferences, achieving LLMs' internal alignment. Experimental results across four knowledge-intensive QA datasets demonstrate that DPA-RAG outperforms all baselines and seamlessly integrates both black-box and open-sourced LLM readers. Further qualitative analysis and discussions provide empirical guidance for achieving reliable RAG systems. Our code and example dataset are available at https://anonymous.4open.science/r/dpa-rag-anonymous-B7B7.

## CCS Concepts

• **Information systems** → **Retrieval models and ranking**.

## Keywords

Retrieval-Augmented Generation, Large Language Model, Alignment

**ACM Reference Format:**

Anonymous Author(s). 2025. Understand What LLM Needs: Dual Preference Alignment for Retrieval-Augmented Generation. In *Proceedings of The 2025 ACM Web Conference (The Web Conference '25)*. ACM, New York, NY, USA, 29 pages. https://doi.org/XXXXXXX.XXXXXXX

## 1 Introduction

The emergence of large language models (LLMs) [1, 66, 68, 106] has profoundly revolutionized a variety of real-world tasks expressed in natural languages [11, 48, 51, 53, 101, 116]. However, when faced

with knowledge-intensive tasks, relying solely on internal knowledge for reasoning may easily expose LLMs to factual inconsistency and hallucination [7, 121]. To alleviate these issues, researchers use retrieval-augmented technology [17, 35] to assist LLMs in integrating relevant knowledge from the web (e.g. Wikipedia [95]) or other external knowledge bases, providing a promising solution to improve the quality of generated answers [73]. [1]

In an ideal retrieval-augmented generation (RAG) system, the goal is to enhance LLMs by incorporating supporting documents that align with their intrinsic knowledge preferences, thus facilitating reasoning. However, in practical applications, the retriever and the LLM-based reader serve as separate components within the RAG system, each with distinct model architectures, training objectives, and task formats [6, 38]. These differences often result in documents retrieved by vector similarity failing to meet the specific knowledge demands for LLM reasoning. Moreover, retrieved documents could even conflict with the self-knowledge of LLMs, potentially disrupting LLMs' original reasoning abilities [60, 71].

As depicted in Figure 1, we perform a preliminary analysis on GPT-3.5 on three QA benchmarks, which compares two setups: LLM answering questions directly and answering questions by referencing different types of retrieved documents. We could categorize results into four distinct conditions: (1) **Both Correct**: the question can be resolved directly by the LLM or through the retrieved documents. (2) **Aligned Knowledge**: LLM gives the wrong answer, but the retrieved documents guide LLM to provide the right solution. (3) **Unaligned Knowledge**: LLM gives the right answer, but the retrieved documents may mislead it. (4) **Both Incorrect**: neither the retrieved documents nor the LLM can provide an answer correctly. Then we have the following observations: in the scenario of aligned knowledge, it is notable that documents with low vector similarity (e.g., ranked 100th) still support the LLM in deducing correct answers. Conversely, within the unaligned knowledge scenario, several documents with high vector similarities tend to mislead LLM more than those with lower similarities (*e.g.*, 10th vs 100th). Surprisingly, even some documents that contain relevant grounding information struggle to align with the LLM's preferences [34]. These results highlight our statement that "The retrieved documents do not exactly match the knowledge required for LLM reasoning". Therefore, mitigating the preference gap between the LLM and the retriever emerges as a critical challenge in developing a reliable RAG system.

To address the above limitation, we propose a **D**ual **P**reference **A**lignment for **R**etrieval-**A**ugmented **G**eneration (**DPA-RAG**), a universal framework designed to align diverse preference knowledge within RAG systems. DPA-RAG consists of three key components: (1) **Preference Knowledge Construction**: motivated by our preliminary results in Figure 1, we first extract the specific knowledge that significantly affects LLMs' reasoning preferences.

---
[1]

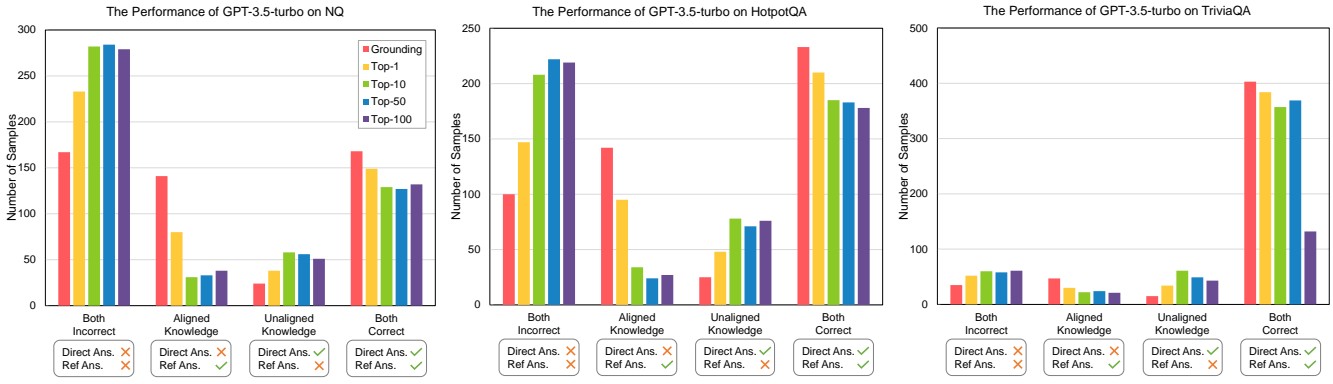

**Figure 1: The results for GPT-3.5 comparing direct responses and answers referencing different retrieved documents (Grounding, 1st, 10th, 50th, 100th) on three QA benchmarks.**

Then we introduce five query augmentation strategies and a quality filtering process to synthesize high-quality preference knowledge. (2) **Reranker-LLM Alignment**: To meet the diverse knowledge preferences of LLMs, we carefully design multi-grained alignment tasks for fine-tuning a preference-aligned reranker. Specifically, we jointly integrate pair-wise, point-wise, and contrastive preference alignment abilities into the reranker via multi-task optimization [83]. By this means, the reranker could provide the necessary knowledge for LLM's inference, achieving external alignment between the retriever and the LLM. (3) **LLM Self-Alignment**: To further enable LLMs to concentrate on knowledge aligned with their reasoning preferences, we introduce a pre-aligned phrase before the vanilla SFT stage. This stage allows the LLM to capture preference-aligned knowledge from multiple documents, completing the LLM's internal self-alignment.

To summarize, our contributions are as follows:

• Based on our quantitative analysis of GPT-3.5 across three QA benchmarks, we reveal the inherent preference gaps between the retriever and the LLM-based reader in RAG systems.

• We propose DPA-RAG, a universal framework designed to align the diverse knowledge preferences of LLMs within RAG systems. DPA-RAG achieves dual preference alignment in two aspects: (1) It jointly integrates multi-grained preference alignment abilities into the reranker, facilitating external alignment across RAG components. (2) It introduces a pre-aligned phrase prior to the standard SFT stage, guiding LLMs to concentrate on the aligned knowledge, thereby unlocking the internal alignment abilities of the LLMs.

• To overcome the scarcity and limited diversity of preference data, we devise five novel query augmentation strategies and a quality filtering process, aiming at automatically synthesizing high-quality preference data for effectively aligning downstream models.

• Experimental results on four knowledge-intensive QA datasets demonstrate the effectiveness of DPA-RAG. Further analysis across dimensions such as model parameters, preference alignment, data quality, and training strategies confirm DPA-RAG's role as a plug-and-play solution, providing practical insights for developing reliable RAG systems.

## 2 Related Work

**Preference Alignment for Large Language Models.** Traditional Preference alignment (PA) methodologies [16, 21, 23, 98] are designed to tailor pre-trained language models to reflect human preferences. Recently, a series of works have relied on reinforcement learning (RL) [82] to align LLMs with human preferences [68]. Owing to the sensitivity of RL's parameters and the complex process of reward modeling, research works [43, 45, 46, 61, 85, 88, 104, 115, 122] represented by DPO [76] further tried to optimize the loss function and reward scoring mechanism for pruning. However, depending on annotations from humans or expert models still increases the alignment cost. To construct reliable RAG systems, a branch of studies [8, 20, 87] aims to align the retriever with supervision signals generated by LLMs, showcasing remarkable alignment potential. Conversely, other studies attempt to improve the alignment abilities of RAG systems by implementing a multi-round retrieval paradigm [24, 78, 93, 96, 108, 125] and filtering out noise from the training corpus [26, 99, 100, 119, 120]. These approaches, however, often suffer from a lack of multi-level alignments, which limits their ability to adapt to the diverse knowledge preferences of LLMs. In our paper, we introduce DPA-RAG, a system that bridges this gap by aligning the retriever to adapt to the diverse knowledge preferences of LLMs without relying on external expert annotations.

**Reranking Techniques for Retrieval Augmented Generation.** In the RAG system, the reranker is designed to rank a list of retrieved documents to accurately meet LLMs' demands. A series of sentence transformer models [31, 64, 77, 105] have achieved excellent fine-grained ranking by better aligning the representations between queries and documents. With the rapid development of prompt learning [44], point-wise generative reranking frameworks [28, 63, 72, 126] have transformed traditional discriminative tasks into a Seq2seq paradigm, showcasing promising initial alignment abilities. The recent development and application of LLMs have introduced innovative pair-wise and list-wise rerankers, such as RankGPT [90], PRP [74], LRL [57] and RankLLaMA [56]. These models have brought multi-perspectives in addressing the fine-grained reranking problem. Moreover, in response to the unique

preferences of different users, various methods [40, 58, 70, 80, 86] have been developed to achieve personalized user sorting, yielding significant results in aligning with industrial scenarios. These advancements inspire us to distill the preferences of LLMs into the reranker, facilitating effective alignment between the RAG system's components.

## 3 Methodology

To address the misalignment between different components of retrieval-augmented generation (RAG) and improve overall generation performance, we propose the DPA-RAG framework, which is illustrated in Figure 2. In general, DPA-RAG improves traditional RAG architecture in two main aspects: (1) we fine-tune a preference-aligned reranker between the retriever and the LLM to selectively filter out knowledge that aligns with LLMs' knowledge preferences (§3.3), and (2) we design a self-alignment mechanism that fine-tunes the LLM to better recognize and utilize knowledge consistent with its reasoning preferences (§3.4). To acquire the LLM's preference knowledge, we devise a three-step data construction method, motivated by our preliminary analysis of how different types of retrieved documents affect RAG performance (§3.2). Below, we will first introduce the task definition (§3.1) and then delve into the specifics of our approach.

### 3.1 Task Definition

Compared to standard text generation, RAG often follows a *retrieve-then-read* paradigm [35], where an additional retriever is introduced to collect external knowledge and enhance the generation process. This architecture involves constructing a *query $q$* to reflect the information needs of the generation. For example, in question-answering systems, the input question is often used as the query. Given the query $q$, the retriever $R$ returns relevant documents from a corpus $D_q = \{d_i\}_{i=1}^{N}$ with $N$ documents. The relevance between document $d$ and the query $q$ can be measured by various methods. In this work, we employ a dense retriever that utilizes dual encoders to obtain hidden representations for both the query and the documents. The relevance score is then calculated by computing the dot-product similarity between these representations, enabling the retrieval of the top-$k$ documents $D_{\text{retrieve}}$:

$$D_{\text{retrieve}} = \text{argtop-}k \left[ E_{\text{d}}(d_i)^\top \cdot E_{\text{q}}(q) \mid i = \{1 \dots N\} \right]. \quad (1)$$

While the retrieved documents are relevant to the query, they may not necessarily contain the knowledge required by the LLMs. Therefore, in this study, we introduce a reranker $E_r$ to rerank $D_{\text{retrieve}}$ and filter out the documents $D_{\text{rerank}}$, which include only those documents aligned with the LLMs' preferences, *i.e.*, $D_{\text{rerank}} = E_r(q, D_{\text{retrieve}})$. Finally, the LLMs read from the reranked documents and generate the target text based on the query:

$$y = \text{LLM}(q, D_{\text{rerank}}) = \log P_\theta (q, D_{\text{rerank}}), \quad (2)$$

where $P_\theta$ represents the LLM's generation probability distribution.

Recognizing that LLMs might struggle to effectively utilize retrieved knowledge, we also design a self-alignment mechanism to optimize $\theta$ for RAG tasks.

## 3.2 Preference Knowledge Construction

To mitigate the misalignment between different RAG components, a critical step is to collect data that reflects LLMs' knowledge preferences. Therefore, we design a three-step method to gradually mine, augment, and filter out high-quality preference knowledge of LLMs, which are shown in the Figure 2.

**Preference Knowledge Extraction.**    To align with LLMs' knowledge preferences, it is essential to identify the specific knowledge that can bring performance gains or harms during the model's inference process.

Motivated by the preliminary analysis in Figure 1, given the training set $\widetilde{D}_{\text{train}} = \{q_i, D_{q_i}, y_{q_i}\}_{i=1}^{N_{\text{train}}}$, where each sample includes a query $q_i$, top-$k$ retrieved documents $D_{q_i} = \{d_i\}_{i=1}^{k}$ and an answer $y_{q_i}$. We guide LLMs to directly answer questions or response by referencing different types of documents, aiming to filter out samples from $\widetilde{D}_{\text{train}}$ that reflects LLMs' knowledge preferences.

To ensure the distinctiveness among these documents, we hierarchically sample four documents from $D_{q_i}$ to construct the document subset $D_{q_i}^{\text{sub}} = \{d_i \mid i = 1, 25, 50, 100\}$ for each query, as shown in the upper part of Figure 2. Consequently, we also categorize the results of LLMs into *"Both Correct"*, *"Both Incorrect"*, *"Aligned Knowledge"*, and *"Unaligned Knowledge"*. From $\widetilde{D}_{\text{train}}$, we selectively extract samples whose document subsets $D_{q_n}^{\text{sub}}$ contain at least one document labeled as *"Aligned Knowledge"* or *"Unaligned Knowledge"*. This allows us to obtain the preference dataset $\widetilde{D}_{\text{pref}} = \{qi, D_{q_i}^{\text{sub}}, Y_i^{\text{sub}}\}_{i=1}^{N}$, where $Y_i^{\text{sub}} = \{y_i \mid i = 1, 25, 50, 100\}$ denotes the preference labels of $D_{q_i}^{\text{sub}}$, corresponding to the four distinct categories[2].

The motivation behind this selection process is that documents labeled as *"Aligned Knowledge"* or *"Unaligned Knowledge"* provide the LLM with a clear positive or negative impact during reasoning. Due to the difficulty in distinguishing the role of retrieved documents labeled as *"Both Correct"* or *"Both Incorrect"*, we choose to discard them.

**Diverse Query Augmentation.**    After obtaining the dataset $\widetilde{D}$pref that reflects the preferences of the LLM, we encountered an issue: data scarcity — $\widetilde{D}$pref contains only 20% of the data from $\widetilde{D}_{\text{train}}$. This scarcity hinders subsequent fine-tuning and alignment of the LLM. Furthermore, data sparsity leads to limited patterns, which in turn results in insufficient diversity and complexity in the data [47, 118]. To address these limitations, we draw inspiration from various augmentation techniques [37, 51, 53, 112, 116] and propose five query augmentation strategies specifically designed for the RAG system:[3]

• **Rephrasing.** Rephrase the original query with the same intention.

• **Complexity.** Increase the semantic complexity of the original query.

• **Decomposition.** Decompose the original query into several sub-problems.

• **Constraint.** Add more conditional and constrained statements to the original query.

• **SPARQL.** Rewrite the original query based on the SPARQL syntax and generate it directly.

---

[2]We also design a softer criterion for four conditions in Appendix C.3
[3]Detailed information on the different augmentation strategies can be found in Appendix C.2

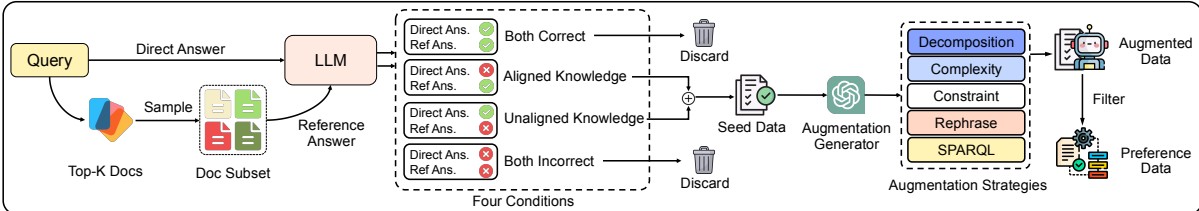

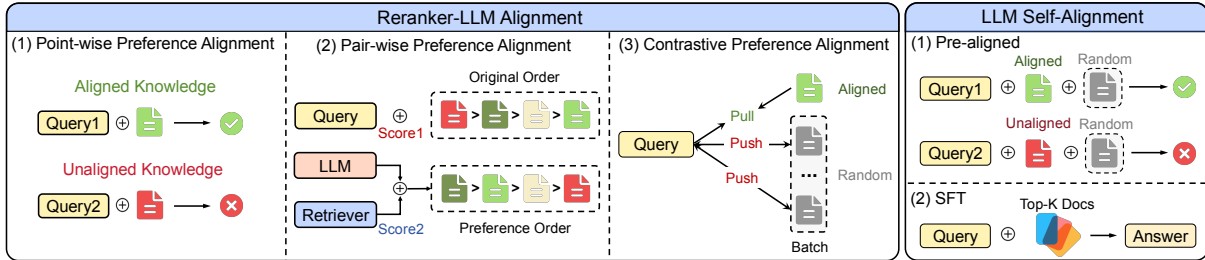

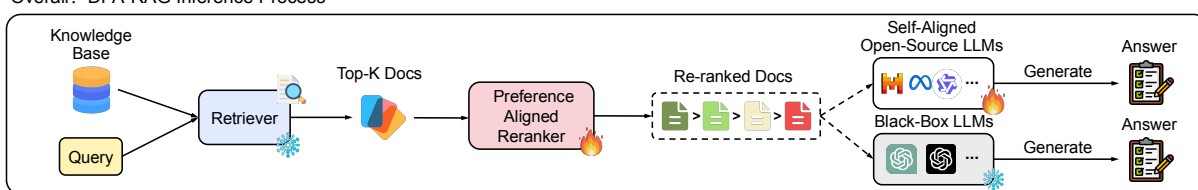

**Figure 2: The overall framework of DPA-RAG. The upper part shows the pipeline of preference knowledge construction. The middle part displays the task format of dual preference alignment. The bottom part illustrates our inference process.**

We utilize GPT-3.5-turbo generate different augmneted datasets $\{\widetilde{D}_{r_i}\}_{i=1}^n$, and then merge them with original dataset $\widetilde{D}_{\text{ori}}$, which can be formulated as $\widetilde{D}_{\text{pref}}^{\text{ori}} = \widetilde{D}_{\text{pref}}^{\text{ori}} \cup (\cup_{i=1}^{n} \widetilde{D}_{r_i})$.

To control the augmented data's quality, we introduce a quality filtering procedure by a natural language inference (NLI) model. Given the original query $q$ as the *"premise"* and the augmented query $q_{\text{aug}}$ as the *"hypothesis"*, the NLI model seeks to determine the semantic relationship between the two queries. The relation can be categorized as *entailment*, *contradiction*, or *neutral*, as follows:

$$p_\theta(\cdot \mid q, q_{\text{aug}}) = \text{softmax}\left(\text{score}_\theta(q, q_{\text{aug}})\right), \tag{3}$$

where $\text{score}_\theta : \mathbb{R}^{k \times \ell_q} \times \mathbb{R}^{k \times \ell_{q_{aug}}} \to \mathbb{R}^3$ is a scoring function dependent on the model's parameters $\theta$. To maintain intent consistency between the original and augmented datasets, we exclude any augmented data labeled as "contradiction" (approximately 20%).

### 3.3 Reranker-LLM Alignment

After obtaining $D_{\text{pref}}$, we introduce multi-grained preference alignment tasks to jointly fine-tune a reranker, aiming to filter retrieved knowledge that aligns with LLM preferences.

**Point-wise Preference Alignment.** Distinguishing beneficial or harmful knowledge of LLMs is essential for aligning their p references. Hence, from each sample $\{q_i, D_{q_i}^{\text{sub}}, Y_i^{\text{sub}}\} \sim \widetilde{D}_{\text{pref}}$, we can further extract one sub-sample $\{q_i, d_i, y_i\}$ where $y_i$ is labeled as *"Aligned Knowledge"* or *"Unaligned Knowledge"*. As shown in Figure

2, we use $\{q_i, d_i, y_i\}_{i=1}^N$ to fine-tune the Reranker model $E_r(\theta)$ with binary cross-entropy loss [84], achieving a point-wise preference alignment:

$$\mathcal{L}_{\text{point}} = -\frac{1}{N} \sum_{i=1}^{N} \left[y_i \log\left(p_\theta(q_i, d_i)\right) + (1 - y_i) \log\left(1 - p_\theta(q_i, d_i)\right)\right],$$

where $y_i$ is label (Postive / Negative) for judging the $d_i$ is aligned or unaligned knowledge.

**Pair-wise Preference Alignment.** Since point-wise alignment empowers the reranker to identify LLM's favored knowledge, enhancing the reranker to prioritize this preferred knowledge presents a new challenge. Therefore, we propose a pair-wise preference ranking task for fine-grained alignment. In detail, given $\{q_i, D_{q_i}^{\text{sub}}, y_i^{\text{sub}}\} \sim \widetilde{D}_{\text{pref}}$, we derive an order $\{o_i\}_{i=1}^K$ of the documents subset $D_{q_i}^{\text{sub}} = \{d_i\}_{i=1}^K$ based on the initial similarity scores from the retriever.

Our idea is elegantly simple: we leverage the LLM within the RAG system as a preference reward model $r_\theta$ to score documents, eliminating the need for external experts. To mitigate bias from relying solely on LLM-generated preference scores [127], we calculate the preference score $s_i$ for each query by weighting both the LLM preference score $r_\theta$ and the original similarity score $s_R(\cdot)$ from the retriever:

$$s_i = a \cdot r_\theta(q, d_i) + (1 - a) \cdot s_R(q, d_i), \tag{4}$$

where $s_i$ denotes the preference score of the $i$-th retrieved document. We then sort the documents according to these preference scores to

obtain the LLM's knowledge preference order $\{\hat{o}_i\}_{i=1}^{K}$. Subsequently, we integrate the preference order into the reranker using RLHF loss [68, 89]:

$$\mathcal{L}_{\text{pair}} = -\frac{1}{C_k^2} \mathbb{E}_{(q,d_w,d_l,y_w,y_l)\sim\widetilde{D}_{\text{pref}}} \tag{5}$$
$$[\log(\sigma(p_\theta(q,d_w,y_w) - p_\theta(q,d_l,y_l)))],$$

where $y_w$ and $y_l$ represent the labels for documents $d_w$ and $d_l$, corresponding to *"winner"* or *"loser"* in the preference order $\{\hat{o}_i\}_{i=1}^{K}$. $p\theta$ denotes the logits of the output.[4]

**Contrastive Preference Alignment.** To align query representations with the LLM's preferred knowledge, we employ contrastive learning [3, 94] to fine-tune our reranker, thereby preventing the LLM from being misled by highly similar but unaligned knowledge. Unlike previous pairwise approaches [76], our $\widetilde{D}$pref dataset associates each query with multiple documents, rather than a single positive or negative example. Considering this one-to-N scenario, we employ Supervised Contrastive Learning (SCL) [32] to fully leverage $\widetilde{D}_{\text{pref}}$. In our task, the query serves as an anchor point $h_q$. Aligned documents are treated as positive samples $h_p$, while documents randomly sampled from other instances in the batch act as negative samples $h_n$. As shown in Figure 2, SCL seeks to reduce the distance of queries and positive samples $h_p$, while increasing the distance from negative samples $h_n$ in the semantic space. The loss $\mathcal{L}_{\text{CPA}}$ is formulated as follows:

$$\mathcal{L}_{\text{CPA}} = -\sum_{i=1}^{N_t} \frac{1}{N_{y_i}-1} \sum_{j=1}^{N_t} \mathbf{1}_{i\neq j}\mathbf{1}_{y_i=y_j} \log \frac{\exp(h_q \cdot h_p/\tau)}{\sum_{k=1}^{N_t} \mathbf{1}_{i\neq k}\exp(h_q \cdot h_n/\tau)},$$

where $N_t$ is the nums of samples in each batch. $N_{y_i}$ denotes samples in the batch with same label as $y_i$. $\tau$ is a temperature parameter. $\mathbf{1}$ is an indicator.

**Multi-task Optimization.** Optimizing multi-grained preference tasks via Multi-task Learning (MTL) [9, 79] offers an efficient way for fine-tuning the reranker. However, learning tasks jointly may further introduce potential bias and conflicts [54]. To tackle this challenge, we employ the MGDA-UB [83], aiming to dynamically find a pareto optimal [41] solution for balancing multi-task optimization.

By utilizing MGDA-UB to optimize the MTL weights $\{c^t\}_{t=1}^{T}$ for $T$ tasks. We finally obtain the multi-grained alignment loss function as:

$$\mathcal{L}_{\text{total}} = c^1\mathcal{L}_{\text{point}} + c^2\mathcal{L}_{\text{pair}} + c^3\mathcal{L}_{\text{CPA}} \tag{6}$$

### 3.4 LLM Self-Alignment

After initially aligning the preferences between external RAG components, in this section, we focus on guiding LLMs to emphasize aligned knowledge during the reasoning process to achieve internal alignment. Inspired by several pre-alignment works [42, 97], we introduce a pre-aligned stage to assist LLMs in implicitly identifying the knowledge crucial for reasoning [26].

**Pre-aligned Stage.** As illustrated in Figure 2, for each sample $\{q_i, D_{q_i}^{\text{sub}}, Y_i^{\text{sub}}\} \sim \widetilde{D}_{\text{pref}}$, we randomly select one document $d_q$ labeled *"Aligned Knowledge"* or *"Unaligned Knowledge"* from $D_{q_i}^{\text{sub}}$,

along with $k-1$ random documents from the retrieved corpus $D = \{d_i\}_{i=1}^{N}$. This selection process constructs a top-$k$ document set $D_{\text{align}} = \{d_q, d_{\text{rand}_1}, \ldots, d_{\text{rand}_{k-1}}\}$ for each query $q$. Then we perform the following training objective with task specific template:[5]

$$\mathcal{L}(\theta) = \sum_{(q_n, D_q, y_n)\in D_{\text{pref}}} \log P_\theta\left(y_n | \text{prompt}(q_n, D_{\text{align}})\right), \tag{7}$$

**Prompt:** Given the documents $\{D_{\text{align}} = (d_q, d_{\text{rand}_1}, \ldots, d_{\text{rand}_{k-1}})\}$. Answer the following question based on the given information or your internal knowledge without the source with a few words. Query: $\{q\}$.
[Judgement]: document-$\{i_{d_q}\}$ is Positive or Negative knowledge for answering question.

where $\log P(\cdot)$ denote probability distribution of LLM's output. $\theta$ denotes model parameters. $\{i_{d_q}\}$ represents the position of the preference document. LLMs will implicitly learn the ability to capture self-preferred knowledge from top-$k$ documents by distinguishing $y \in \{positive, negative\}$ during the pre-aligned task.

**Supervised Fine-tuning Stage.** Following the pre-aligned task, we load pre-trained parameters and perform subsequent Supervised Fine-tuning (SFT) for QA tasks using the same objective described in Equation (7). We utilize the traditional QA format training set $\widetilde{D}_{\text{train}} = \{q_i, D_{q_i}, y_{q_i}\}_{i=1}^{N_{\text{train}}}$. Moreover, we merge five augmented datasets $\{\widetilde{D}_{r_i}\}_{i=1}^{5}$ with $\widetilde{D}_{\text{train}}$. Using the preference-aligned reranker $E_r$, we reorder the documents and filter out the top-k documents as described in Equation (8), forming the final training set $\widetilde{D}_{\text{train}}^{\text{rank}} = \{q_i, D_{q_i}^{\text{rank}}, y_{q_i}\}_{i=1}^{N_{\text{train}}}$ of SFT stage.

$$D_{q_i}^{\text{rank}} = \text{argtop-}k\left[E_r(q_i, D_{q_i})\right] \tag{8}$$

The preference knowledge identification capability developed during the pre-aligned stage enables LLMs to focus more effectively on aligned knowledge during the SFT stage, thereby enhancing their internal alignment potential. The prompt template for the SFT stage is as follows:

**Prompt:** Given the documents $\{$Top-K Docs: $D_q^{\text{rank}}\}$. Answer the following question based on the given information or your internal knowledge without the source with a few words. Query:$\{q\}$.

## 4 Experiments

### 4.1 Datasets and Metrics

We select four question answering (QA) datasets covering three types, including **(1) Open-Domain QA**, represented by NaturalQuestions (NQ) [33] and TriviaQA (TQA) [27]; **(2) Multi-Hop QA**, represented by HotpotQA (HQA) [107]; and **(3) Knowledge Base QA**, represented by WebQuestionsSP (WebQSP) [109]. Table 2 illustrates the statistics of them. For evaluation metrics, we use Hit@1 for the accuracy of the top-ranked response and F1 score to assess the quality and similarity to the ground-truth. We also provide a detailed **estimation of the training and inference FLOPs** of DPA-RAG compared to baselines in Appendix A.3 and Table 5, validating the efficiency of DPA-RAG. More details of the experimental setup are listed in Appendix B.

---

[4]An in-depth discussion on scoring mechanisms for different LLMs can be found in Appendix A.2.

[5]The document $d_q$ is placed at a random position among $k$ documents.

**Table 1: Results of DPA-RAG and different kinds of baselines on four QA benchmarks.**

| Method | Reader | NQ | | Trivia-QA | | Hotpot-QA | | WebQSP | |
|---|---|---|---|---|---|---|---|---|---|
| | | Hit@1 | F1 | Hit@1 | F1 | Hit@1 | F1 | Hit@1 | F1 |
| *Traditional RAG with DPR* | | | | | | | | | |
| RAG [69] | GPT-3.5 | 47.47 | 47.99 | 75.04 | 74.13 | 26.28 | 32.84 | 67.97 | 63.33 |
| RAG [67] | GPT-4 | 54.04 | 51.19 | 79.98 | 76.85 | 28.46 | 33.87 | 71.30 | 67.20 |
| RAG [92] | LLaMA2-7B | 50.94 | 54.76 | 63.90 | 63.80 | 31.40 | 38.90 | 68.52 | 64.22 |
| RAG [92] | LLaMA2-13B | 56.60 | 60.60 | 70.43 | 71.32 | 36.31 | 45.23 | 76.39 | 78.63 |
| RAG [59] | LLaMA3-8B | 54.81 | 58.33 | 69.54 | 71.21 | 34.28 | 42.29 | 72.82 | 73.94 |
| RAG [4] | Qwen2-7B | 52.01 | 56.13 | 63.88 | 66.52 | 31.39 | 39.70 | 75.98 | 77.82 |
| *RAG with DPR & Reranker* | | | | | | | | | |
| RAG+RankGPT [90] | LLaMA2-7B | 47.81 | 52.37 | 59.05 | 56.39 | 28.32 | 37.06 | 66.32 | 62.22 |
| RAG+LRL [57] | LLaMA2-7B | 48.09 | 53.06 | 60.33 | 56.86 | 29.13 | 37.81 | 67.43 | 63.44 |
| RAG+PRP [74] | LLaMA2-7B | 51.91 | 56.17 | 62.28 | 57.98 | 31.90 | 40.87 | 68.54 | 64.08 |
| RAG+RankLLaMA [56] | LLaMA2-7B | 52.18 | 56.62 | 62.34 | 58.05 | 32.31 | 41.39 | 69.11 | 65.70 |
| RAG+BGE [105] | LLaMA2-7B | 52.43 | 56.92 | 62.70 | 57.58 | 32.53 | 41.73 | 70.20 | 68.80 |
| RAG+BCEmbedding [62] | LLaMA2-7B | 49.91 | 53.19 | 61.93 | 57.67 | 31.52 | 40.59 | 68.20 | 65.40 |
| RAG+ColBERTv2 [81] | LLaMA2-7B | 51.49 | 56.02 | 62.34 | 58.16 | 31.72 | 40.79 | 69.70 | 66.90 |
| *Preference-aligned Methods for RAG* | | | | | | | | | |
| REPLUG [87] | GPT-3.5 | 49.67 | 50.58 | 75.67 | 75.34 | 27.30 | 34.30 | 69.59 | 66.22 |
| RA-Judgement [78] | GPT-3.5 | 48.52 | 50.18 | 76.21 | 76.58 | 26.50 | 32.81 | 66.07 | 68.32 |
| KnowPAT [120] | LLaMA2-7B | 51.42 | 54.82 | 63.20 | 65.20 | 29.00 | 37.40 | 68.73 | 65.31 |
| RRHF [117] | LLaMA2-7B | 50.11 | 52.01 | 62.50 | 60.20 | 28.16 | 35.40 | 66.90 | 63.10 |
| RAFT [119] | LLaMA2-7B | 50.24 | 53.86 | 60.10 | 57.40 | 30.20 | 35.80 | - | - |
| FILCO [99] | LLaMA2-7B | 52.71 | 55.32 | 67.30 | 67.80 | 32.70 | 40.80 | 69.96 | 68.34 |
| *Our Method: DPA-RAG* | | | | | | | | | |
| DPA-RAG | GPT-3.5 | 51.60 (+4.13) | 52.80 (+4.81) | 78.65 (+3.61) | 77.05 (+2.92) | 28.42 (+2.14) | 36.12 (+3.28) | 71.80 (+3.83) | 69.20 (+5.87) |
| DPA-RAG | GPT-4 | 56.45 (+2.41) | 53.28 (+2.09) | 84.41 (+4.43) | 80.08 (+3.23) | 33.79 (+5.33) | 37.67 (+3.80) | 73.12 (+1.82) | 74.83 (+7.63) |
| DPA-RAG | LLaMA2-7B | 56.03 (+5.09) | 60.19 (+5.43) | 70.16 (+6.26) | 70.29 (+6.49) | 35.23 (+3.83) | 43.34 (+4.44) | 72.40 (+3.88) | 71.80 (+7.58) |
| DPA-RAG | LLaMA2-13B | 59.19 (+2.59) | 62.97 (+2.37) | 74.18 (+3.75) | 75.53 (+4.31) | 41.07 (+4.76) | 49.60 (+4.37) | 80.28 (+3.89) | 81.74 (+3.11) |
| DPA-RAG | LLaMA3-8B | 57.43 (+2.62) | 61.02 (+2.69) | 72.04 (+2.50) | 73.58 (+2.37) | 36.01 (+1.73) | 44.32 (+2.03) | 74.26 (+1.44) | 76.11 (+2.17) |
| DPA-RAG | Qwen2-7B | 54.66 (+2.65) | 58.84 (+2.71) | 68.58 (+4.70) | 70.26 (+3.74) | 34.56 (+2.87) | 42.47 (+2.77) | 78.66 (+2.68) | 80.53 (+2.71) |

**Table 2: Statistics for the QA datasets.**

| Dataset | # Training | # Development | # Test |
|---|---|---|---|
| NQ | 79,200 | 8,700 | 3,600 |
| TriviaQA | 78,800 | 8,800 | 11,300 |
| HotpotQA | 88,900 | 5,600 | 5,600 |
| WebQSP | 2,840 | 250 | 1,600 |

**Table 3: Ablation study on NQ and TQA.**

| Method | NQ | | TQA | |
|---|---|---|---|---|
| | Hits@1 | F1 | Hits@1 | F1 |
| Standard RAG | 50.94 | 54.76 | 63.90 | 63.80 |
| DPA-RAG | 56.03 | 60.19 | 70.16 | 70.29 |
| *w/o* PA-Rerank | 52.80 | 60.19 | 66.26 | 66.39 |
| *w/o* Pre-Align | 54.31 | 58.95 | 61.69 | 61.35 |
| *w/o* Pre-Align + PA-Rerank | 51.91 | 55.98 | 58.24 | 59.30 |
| *w/o* Query Augmentation | 54.81 | 57.45 | 61.28 | 60.93 |

## 4.2 Main Results

The experimental results are shown in Table 1. In general, our DPA-RAG significantly outperforms all baselines across four datasets in different setups. This highlights the superiority of our approach. We further have the following observations:

(1) Compared to traditional RAG baselines, DPA-RAG (LLaMA2-7B) shows a remarkable performance improvement (over 5%) across all four datasets. More importantly, this improvement is consistent across various models, including LLaMA2-13B, Qwen2-7B, LLaMA3-8B, GPT-3.5, and GPT-4. This indicates the broad applicability and generalizability of our method.

(2) For reranker-based methods, we find that smaller rerankers such as BGE and ColBERTv2 can achieve comparable or even better performance than LLM-based rerankers. This result validates our motivation for using BGE as the alignment backbone, as it combines efficiency with effectiveness.

(3) Among preference-aligned methods, DPA-RAG outperforms direct alignment methods (*i.e.*, REPLUG and RA-Judgement), which rely on logits. This emphasizes the value of implementing multi-grained alignments within our framework. Surprisingly, Filco, which

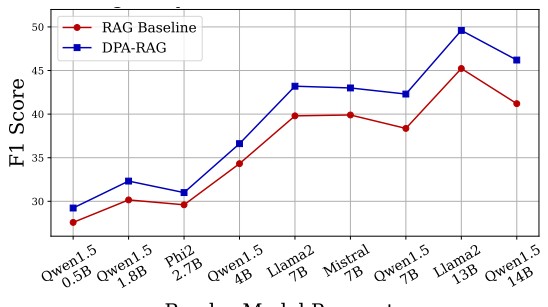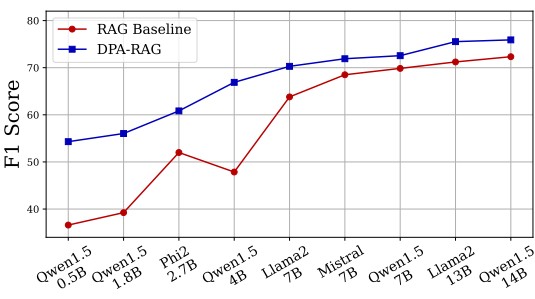

Figure 3: The scaling analysis of different parameter scales for HQA (left) and TQA (right).

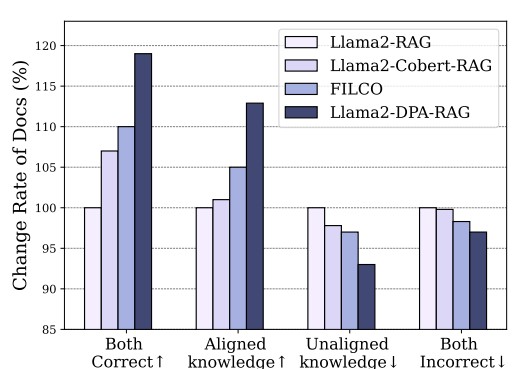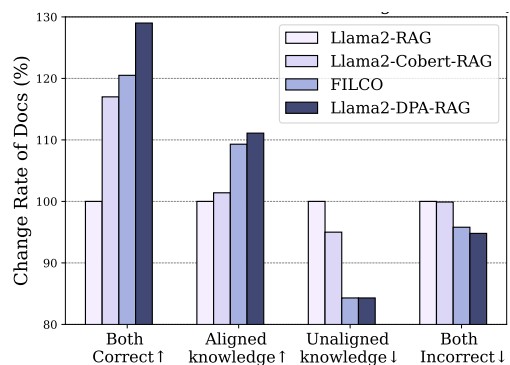

Figure 4: The comparison experiment of preference alignment on NQ, TQA.

employs data filtering, shows robust alignment capabilities, confirming that unaligned knowledge exists in training corpora. This observation highlights again the importance of our preference optimization at the data level, ensuring that the retrieved and used knowledge is highly relevant and aligned with the LLM's needs. We also supplement more results on Noise Robustness Methods and other baselines in Appendix C.6 and C.7.

**Ablation Study.** To explore the roles of different modules in DPA-RAG. We perform an ablation study and Table 3 shows the results. We use *w/o* to indicate the version *without* a particular module. We can see: (1) The performance of DPA-RAG declines when any component is removed, which suggests that all the components are very effective. (2) Removing the preference aligned reranker (PA-Rerank.) leads to the largest performance drop, indicating a clear knowledge preference gap between RAG components and LLMs. This confirms the benefit of using a preference-aligned reranker for external alignment. (3) The combined performance gains of preference aligned reranker and pre-aligned task are lower than the complete DPA-RAG framework, which implies that integrating both alignment methods yields a mutually reinforcing effect, demonstrating the superiority of our dual alignment strategies. More detailed results can be found in Appendix C.1.

### 4.3 Quantitative Analysis

**Scaling Analysis for Different Model Parameters.** To investigate the impact of parameter scale and RAG performance, we gradually increase the parameters of LLM readers (ranging from

500M to 13B) and evaluate their performance. According to the results in Figure 3, we have following observations:

(1) **Emergence of RAG Capabilities at Lower Parameter Scales (<7B)**: We notice a significant improvement in RAG baseline performance, which sharply rises from 500M to 7B parameters (40% F1 score increase), then stabilizes for parameters beyond 7B. A similar pattern is observed in HQA, indicating a strong correlation between the emergence of RAG capabilities and model parameters. This finding presents an interesting parallel to those reported in LIMA [124], where parameter increases below a certain threshold significantly boost model capabilities.

(2) **Stable Performance Gains with DPA-RAG as Parameters Increase**: Compared to the baseline, DPA-RAG delivers stable improvements as parameter size expands across both datasets, displaying a smoother performance curve.

(3) **Greater Benefits from DPA-RAG in Datasets with More Unalignment**: The performance gains from DPA-RAG exhibit interesting variations between TQA and HQA as parameters increase. In TQA, where the average F1 score is over 60, the model quickly reaches a high-performance threshold as parameters increase, leaving limited room for further improvements through preference alignment. Conversely, HQA, characterized by more extensive unaligned knowledge and a lower average F1 score (below 50), shows that the alignment gains provided by DPA-RAG exceed those from increasing foundational RAG capabilities alone, leading to more improvement in alignment for RAG.

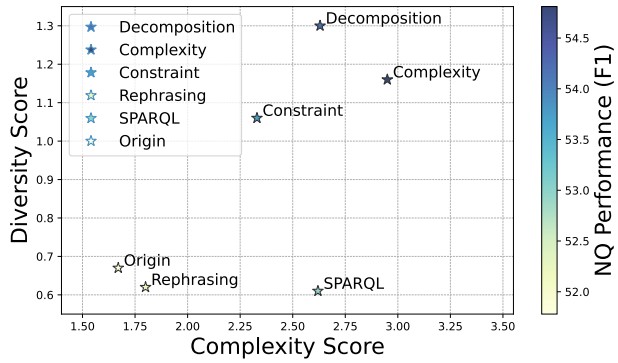

Figure 5: The visualization of different data complexity and diversity.

Table 4: The performance result correlates with complexity and diversity on NQ.

| Aug-Type | Complexity | Diversity | Total | NQ |
|---|---|---|---|---|
| Origin | 1.61 | 0.35 | 1.96 | 51.78 |
| Rephras. | 1.64 | 0.39 | 2.03 | 52.27 |
| SPARQL | 1.77 | 0.39 | 2.16 | 52.95 |
| Constraint | 1.72 | 0.47 | 2.19 | 53.75 |
| Decompos. | 1.77 | 0.51 | 2.28 | 54.16 |
| Complexity | 1.85 | 0.48 | 2.33 | 54.81 |

**Effectiveness on Preference Alignment.** To delve deeper into the impact of preference alignment, in line with the setup in Section 3.2, we conduct a comparative experiment on direct query answering versus referencing top-3 documents. As shown in Figure 4, DPA-RAG consistently achieve the highest scores in the *"Aligned Knowledge"* category across all three datasets, while significantly reducing the *"Unaligned Knowledge"* category. This demonstrates that DPA-RAG effectively aligns retrieved knowledge with the LLM's inherent preferences. Interestingly, the improvement of DPA-RAG in the "Both Correct" category even outperforms that observed in *"Aligned Knowledge"*. Given the significant decrease in *"Unaligned Knowledge"*, this suggests that DPA-RAG prioritizes addressing the conflicts present in retrieved documents. This behavior is in line with our pipeline's core principle: the preference-aligned reranker first externally eliminates misaligned knowledge, and the subsequent self-alignment stage allows the LLM to more effectively and implicitly capture information that is aligned with its preferences. In Appendix C.4, we further verify that the preferences of different LLMs are transferable.

**Discussion on Query Augmentations.** Liu et al. [47] and Lu et al. [50] highlight the significant impact of dataset complexity and diversity on model alignment. To investigate how the complexity and diversity of our augmented queries affect RAG performance, we randomly select 1,000 samples from each dataset and employ Intag technology [50] for automated intent annotation. For each dataset, we measure diversity by calculating $\frac{\text{number of unique tags}}{\text{number of all samples}}$ and complexity by $\frac{\text{number of all tags}}{\text{number of all samples}}$. Figure 5 visualizes the quality of the augmented data, showing that our five methods consistently

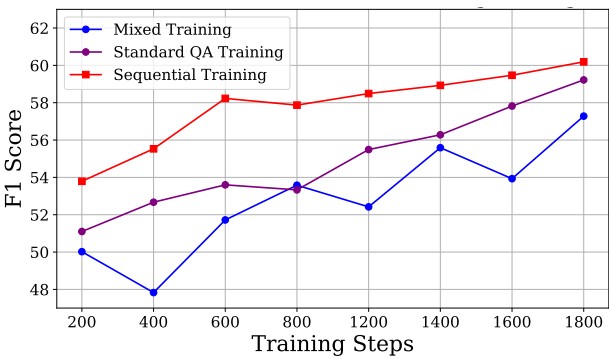

Figure 6: The performance of different training strategies on NQ.

enhance data complexity. Specifically, *Complexity* and *Decomposition* markedly boost both complexity and diversity scores, which also align with the case studies presented in Table 7. Moreover, we mix the augmented data with the original training set in actual proportions and calculate the data quality. Table 4 (left) shows that all five augmentation strategies enhance the LLM's performance to different degrees. Surprisingly, when we sum up the two metrics, the overall trend of performance on NQ increases along with the growth of the total quality score. This insight further validates that in RAG tasks, the effectiveness of query augmentations is highly correlated with their complexity and diversity.

**Sequential Training vs. Mixed Training.** In Section 3.4, we design a knowledge self-alignment task during the pre-aligned phase and further perform sequential SFT on the QA dataset. An alternative approach is directly mixing preference data with QA task data for joint training. Figure 6 illustrates the performance of these two training strategies across training steps. Compared to standard QA fine-tuning, we notice that mixing training data from both tasks leads to a noticeable performance decline and fluctuations. This result may stem from optimization conflicts in multi-task training [14]. However, the sequential training after the pre-aligned phase yields stable performance gains, validating its efficacy. Similar conclusions have been reported in studies on reasoning [15, 91, 97, 103].

## 5 Conclusion

In this paper, we reveal the inherent preference gap among RAG components and first propose DPA-RAG to align diverse knowledge preferences. Specifically, we gradually extract and filter out the LLM preferred knowledge from training set, and propose five high-quality query augmentation strategies to alleviate data sparsity issues. Based on preference data, we jointly integrate pair-wise, point-wise, and contrastive preference alignment abilities into the reranker, achieving external preference alignment among RAG components. Further, we introduce LLM Self-Alignment task to remove knowledge biases and achieve internal alignment. Experimental results demonstrate that DPA-RAG outperforms all strong baselines across four knowledge-intensive QA datasets. Further analysis provides practical insights for developing reliable RAG systems.

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

# Appendix

CONTENTS

## A More Details about DPA-RAG

### A.1 The Overall Algorithm Workflow of DPA-RAG

In this section, we delve into the overall workflow of the DPA-RAG algorithm, which can be divided into **Reranker Training Algorithm** and **LLM-based Generator Training**.

**Reranker Training Algorithm:** Given the train set $\widetilde{D}_{\text{train}} = \{q_i, D_{q_i}, y_{q_i}\}_{i=1}^{N_{\text{train}}}$, we initially perform preference knowledge mining techniques to select, augment and filter the data to construct a preference-aligned dataset $\widetilde{D}_{\text{pref}}$. Subsequently, relying on the $\widetilde{D}_{\text{pref}}$, we perform multi-grained distillation alignments with MGDA-UB

stategy to better fine-tune a preference-aligned reranker. The detailed process is listed in algorithm diagram 1.

**LLM-based Reader Training Algorithm:** As shown in algorithm diagram 2, for open-source LLM-based reader, we directly utilize the preference-aligned reranker to perform preference-based reranking on retrieved documents in $\widetilde{D}_{\text{train}}$[6] and $\widetilde{D}_{\text{test}}$, resulting in sorted datasets $\widetilde{D}_{\text{train}}^{\text{rank}}$ and $\widetilde{D}_{\text{test}}^{\text{rank}}$. In addition, we also construct a dataset $\widetilde{D}_{\text{train}}^{\text{PA}}$ for the knowledge self-alignment task based on $\widetilde{D}_{\text{pref}}$. Initially, we use $\widetilde{D}_{\text{train}}^{\text{PA}}$ for the pre-aligned task, then we load the pre-trained model parameters and then conduct vanilla QA supervised fine-tuning based on $\widetilde{D}_{\text{train}}^{\text{rank}}$. During the inference phase, we input the preference-sorted test set $\widetilde{D}_{\text{test}}^{rank}$ into the LLM to complete the prediction.

For close-source LLM-based reader, the process is more simple: the preference-aligned reranker is used to sort documents in the test set $\widetilde{D}_{\text{test}} \rightarrow \widetilde{D}_{\text{test}}^{\text{rank}}$, then we use LLMs for the prediction process.

### A.2 Preference Scoring Mechanism for Different LLMs

In practice, we find that models with fewer than 7B parameters struggle with instruction-following capabilities, making it difficult for them to perform the scoring task. To address this, we follow the RankLLaMA [56] and RePLUG [87], utilizing the output's logit as the basis for scoring as follow:

$$r_\theta(q, d_i) = \log \mathbf{P}_\theta \left( \text{ prompt } (q, d_i) \right) \tag{9}$$

$$s_i = a \cdot r_\theta(q, p_i) + (1 - a) \cdot s_R(q, p_i) \tag{10}$$

where $q$, $d_i$ denotes the query and top i-th document. $\log \mathbf{P}(\cdot)$ represents the model's probability distribution. Prompt denotes the prompt template. $s_i$ is the final preference score of i-th retrieved document. For the hyper-parameter $a$, we follow QLM Reranker [127] and set it to 0.8 without performing any grid search. Next, we rank them to obtain the preference order $\{o_1, o_2, .., o_n \mid r_\theta, s_R\}$ according to $\{s_i\}_{i=1}^K$.

For the 7B and 13B models, we observe that these models fundamentally possess the capability to follow instructions in our preliminary experiments. Therefore, we prompt them to perform preference scoring from 1 to 5. Then we normalize the preference score $r_\theta(q, d_i)$ and sum it with the retriever's similarity score $s_R(q, d_i)$ as equation 10. Finally, we rank them to obtain the preference order.

As the result in Table 1, for powerful LLMs (such as GPT-3.5 and GPT-4), we find that pair-wise comparative ranking can achieve a more precise preference ordering compared to ranking by scoring each paragraph individually. Therefore, we perform $C_k^2$ pair-wise comparisons of knowledge documents as PRP [74] through LLMs to obtain the preference ordering results.

### A.3 Estimating FLOP of Training and Inference

**Training Budget.** We mainly follow the notations of Scaling Laws here [29]. For each input sample of length in SFT dataset (NQ, TQ, HQ, WebQSP), we can split it into 3 parts:

$$n_{ctx} = n_Q + n_{Docs} + n_R \tag{11}$$

---

[6] The training set $\widetilde{D}_{\text{train}}$ consists of the original training set $\widetilde{D}_{\text{train}}^{ori}$ and $\widetilde{D}_{\text{aug}} \in \widetilde{D}_{\text{pref}}$ with five query augmentations.

---

**Algorithm 1** Reranker Training

---

1: **procedure** CONSTRUCTPREFERENCEDATASET($\widetilde{D}_{\text{train}}$).
2:    $\widetilde{D}_{\text{pref}} \leftarrow \emptyset$
3:    From $(q_i, D_{q_i}, y_{q_i}) \in \widetilde{D}_{\text{train}}$, we select the $\widetilde{D}_{\text{sub}} = \{q_i, D_{q_i}^{\text{sub}}, Y_i^{\text{sub}}\}_{i=1}^N$.
4:    **for all** $\{q_i, D_{q_i}^{\text{sub}}, Y_i^{\text{sub}}\} \in \widetilde{D}_{\text{sub}}$ **do**         ▷ Mine Preference Knowledge
5:       **for all** $\{d_i | i = 1, 25, 50, 100\} \in D_{q_i}^{\text{sub}}$ **do**
6:          $a_{\text{LLM}} \leftarrow$ LLM answer to query $q_i$
7:          $a_{\text{docs}} \leftarrow$ Correct answer from $d_i$
8:          **if** $a_{\text{LLM}} \neq y_n$ and $a_{\text{docs}} = y_n$ **then**
9:             $\widetilde{D}_{\text{pref}} \leftarrow \widetilde{D}_{\text{pref}} \cup \{(q_i, D_{q_i}^{\text{sub}}, Y_i^{\text{sub}})\}$         ▷ Aligned Knowledge
10:             Continue
11:          **else if** $a_{\text{LLM}} = y_n$ and $a_{\text{docs}} \neq y_n$ **then**
12:             $\widetilde{D}_{\text{pref}} \leftarrow \widetilde{D}_{\text{pref}} \cup \{(q_i, D_{q_i}^{\text{sub}}, Y_i^{\text{sub}})\}$       ▷ Unaligned Knowledge
13:             Continue
14:          **end if**
15:       **end for**
16:    **end for**
17:    $G_\theta \leftarrow$ Augmented query generator
18:    $R \leftarrow$ {Complexity, Constraint, SPARQL, Decomposition, Rephrasing}
19:    **for all** $R_i$ in $R$ **do**
20:       **for all** $(q_i, D_{q_i}) \in \widetilde{D}_{\text{pref}}$ **do**
21:          $q_{\text{aug},i} \leftarrow G_\theta(R_i, q_i, D_{q_i})$
22:          $D_{r_i} \leftarrow D_{r_i} \cup \{(q_{\text{aug},i}, D_{q_i}, y_{q_i})\}$
23:       **end for**
24:       $\widetilde{D}_{\text{pref}} \leftarrow \widetilde{D}_{\text{pref}} \cup \left( \cup_{i=1}^n D_{r_i} \right)$
25:    **end for**
26:    $p_\Theta \leftarrow$ NLI model for quality filtering
27:    **for all** augmented query $q_{\text{aug}}$ in $\widetilde{D}_{\text{pref}}$ **do**
28:       $score_\theta \leftarrow p_\Theta(q, q_{\text{aug}})$
29:       **if** $score_\theta$ is not "entailment" **then**
30:          $\widetilde{D}_{\text{pref}} \leftarrow \widetilde{D}_{\text{pref}} \setminus \{(q_{\text{aug}}, D_{q_i}, y_{q_i})\}$
31:       **end if**
32:    **end for**
33:    **return** $\widetilde{D}_{\text{pref}}$
34: **end procedure**
35: **procedure** MULTIGRAINEDDISTILLATIONALIGNMENT($\widetilde{D}_{\text{pref}}$)
36:    Initialize model parameters $\theta^{sh}, \theta^1, \ldots, \theta^T$
37:    **repeat**
38:       Compute losses $\mathcal{L}_{\text{CPD}}, \mathcal{L}_{\text{FPR}}, \mathcal{L}_{\text{SCA}}$
39:       **procedure** MGDA-UB($\theta^{sh}, \theta^1, \ldots, \theta^T, c^t$)
40:          $\mathbf{Z} \leftarrow \sum_{t=1}^T c^t \nabla_{\theta^{sh}} \hat{\mathcal{L}}^t(\theta^{sh}, \theta^t)$
41:          Optimize MTL weights $\alpha^t$ for Pareto optimal solution
42:          $\mathbf{L} \leftarrow \sum_{t=1}^T c^t \hat{\mathcal{L}}^t(\theta^{sh}, \theta^t)$
43:          **return** $\mathbf{L}$
44:       **end procedure**
45:       Update model parameters $\theta^{sh}, \theta^1, \ldots, \theta^T$ to minimize $\mathbf{L}$
46:    **until** convergence
47:    **return** Optimized parameters $\theta^{sh}, \theta^1, \ldots, \theta^T$
48: **end procedure**

---

**Algorithm 2** LLM-based Reader Training

---

1: **procedure** PRE-ALIGN($\widetilde{D}_{\text{pref}}, k$)
2:     **for all** $\{q_i, D_{\text{pref}}, y_{q_i}\} \in \widetilde{D}_{\text{pref}}$ **do**
3:         Select one document from $D_{\text{pref}}$
4:         Randomly select $k - 1$ documents from $D = \{d_i\}_{i=1}^N$
5:         Construct Top-k document set $D_{\text{align}} = \{d_{\text{pref}}, d_{\text{rand}_1}, \ldots, d_{\text{rand}_{k-1}}\}$
6:         Initialize prompt with the selected documents and query
7:     **end for**
8:     Fine-tune the LLMs with the objective $\mathcal{L}(\theta) = \sum\limits_{(q_i, D_{\text{align}}, y_{q_i}) \in \mathcal{D}} \log \mathbf{P}_\theta \left( y_{q_i} | prompt(q_i, D_{\text{align}}) \right)$
9: **end procedure**
10: **procedure** SUPERVISED FINE-TUNING($\mathcal{D}$, Pre-Aligned Parameters)
11:     Load pre-warmed parameters from PreAligned stage
12:     Merge augmented dataset as $\widetilde{D}_{\text{train}} = \widetilde{D}_{\text{train}} \cup (\cup_{i=1}^n \widetilde{D}_{r_i})$
13:     **for all** $\{q_i, D_{q_i}, y_{q_i}\} \in \widetilde{D}_{\text{train}}$ **do**
14:         $D_{q_i}^{\text{rank}} \leftarrow$ Top-K [ Reranker($q_i, D_{q_i}$) ]
15:         $\widetilde{D}_{\text{train}}^{\text{rank}} \leftarrow \{(q_i, D_{q_i}^{\text{rank}}, y_{q_i})\}$
16:     **end for**
17:     Perform supervised fine-tuning
18: **end procedure**

---

| Methods | Reranker/Filter Model | Reader | Total Training FLOPs | Training Hours | Inference Process |
|---|---|---|---|---|---|
| Standard RAG | - | Llama2 (7B) | $2.49 \times 10^{17}$ (1) | 0.8 (1) | LLM (1) |
| DPA-RAG | BGE (125M) | Llama2 (7B) | $3.48 \times 10^{17}$ (2) | 1.1 (2) | Reranker (BGE) + LLM (2) |
| RAG+BGE | BGE (125M) | Llama2 (7B) | $2.49 \times 10^{17}$ (1) | 0.8 (1) | Reranker (BGE) + LLM (2) |
| RAG+RankLlama | Llama2 (7B) | Llama2 (7B) | $1.9 \times 10^{18}$ (5) | 7.6 (5) | Reranker (Llama2) + LLM (4) |
| KnowPAT | - | Llama2 (7B) | $5.0 \times 10^{17}$ (4) | 2 (4) | LLM (1) |
| FILCO | FlanT5 (3B) | Llama2 (7B) | $3.55 \times 10^{17}$ (3) | 1.2 (3) | Filter model (FlanT5) + LLM (3) |

**Table 5: The statistics of Model Types, FLOPs, GPU hours and Inference Process. The numbers in parentheses represent the ranking of resource consumption from lowest to highest.**

$$C_{\text{train}} \approx 6 N n_{ctx} N_s \qquad (12)$$

where $n_Q$, $n_{Docs}$, $n_R$ denotes the length of query, TopK documents and answers respectively. $N, N_s$ denotes the non-embedding parameters and the numbers of samples, which we refer to Chinchilla for calculations. In NQ dataset, $n_Q \approx 15$, $n_{Docs} \approx 478$ and $n_R \approx 2$. Therefore, We estimate the FLOPs and GPU times on NQ dataset in Table 1 of Rebuttal PDF.

**Inference Budget.** Due to the presence of KV cache computations, it is quite difficult to accurately derive the inference FLOPs of different models. Therefore, we quantified the inference steps required by various baselines, which allowed us to roughly rank their inference costs.

**Analysis.** Following the steps outlined above, we carefully calculate the data size, training FLOPs, training time, and inference costs of different methods in Table 5.

(1) In terms of training budgets, we outperform the KnowPAT, FILCO, and RAG+RankLlama methods, particularly when compared to reranker-based and preference alignment baselines.

(2) For inference, our performance is comparable to the classic RAG+bge reranker-based baseline and significantly exceeds that of other baselines.

These results indicate that the resource expenditure of our dual alignment method is reasonable and does not lead to significant additional resource consumption.

## B   More Details on Experiment Setup

### B.1   Datasets

In this section, we report the detailed information of our 4 datasets, including NaturalQuestions (NQ), TriviaQA (TQA), HotpotQA (HQA), WebQuestionsSP (WebQSP).

**Natural Questions (NQ)** [33] dataset, with its approximately 300,000 real Google searches and corresponding answers from Wikipedia, annotated for detailed context and brief replies, is crucial for developing question-answering systems, enhancing AI's comprehension of natural language.

**TriviaQA (TQA)** [27] serves as a benchmark for QA models, with its extensive set of over 650,000 question-answer pairs sourced

from quizzes and trivia competitions. Each question is linked to supporting documents, presenting a challenge for systems to extract correct information from various subjects, which in turn evaluates their information gathering and language comprehension capabilities.

**HotpotQA (HQA)** [107] dataset comprises 113,000 questions necessitating answers through multi-step logic. It pushes the envelope in AI development by demanding linkage of several documents for inferencing comprehensive answers, aiming to improve AI abilities in complex understanding far exceeding simple fact extraction.

**WebQuestionsSP (WebQSP)** [109] dataset consists of more than 4,700 Google Suggest-derived questions, each associated with a query in SPARQL format that retrieves answers from the Freebase. It is specifically crafted for refining QA systems' semantic parsing skills and their ability to transform natural language into formal database queries, thereby pushing the boundaries of AI in processing and understanding intricate queries from real-life scenarios.

## B.2 Prompt Templates

In the vanilla SFT stage, we follow the template of the RA-Judgement as follow [78]:

---

**Prompt Template of SFT Stage**

Given the documents {Top-K Documents}. Answer the following question based on the given information or your internal knowledge with one or few words without the source. Query: {Query}.

---

For the pre-aligned stage, our prompt template is almost aligned with the SFT stage's template. The only difference is that we add an additional judgment statement that allows the LLMs to distinguish whether the influence of the preference document $d_q$ on answering questions is positive or negative, thereby implicitly learning the ability to distinguish between aligned knowledge and unaligned knowledge. The prompt template is displayed as follow:

---

**Prompt Template of Pre-aligned Stage**

Given the documents $\{D_{\text{align}} = (d_q, d_{\text{rand}_1}, \ldots, d_{\text{rand}_{k-1}})\}$. Answer the following question based on the given information or your internal knowledge with few words without the source. Query: $\{q\}$.
[Judgement] document-$\{i_{d_q}\}$ is Positive or Negative knowledge for answering question.

---

where $d_q$ denotes the preference document that influences the LLM's reasoning results for query $q$. $\{d_{\text{rand}_1}, \ldots, d_{\text{rand}_{k-1}}\}$ denotes $k-1$ random documents from the retrieved corpus $D_{\text{align}}$. Moreover, $i_{d_q}$ denotes the order of $d_q$ in $D_{\text{align}}$.

For data augmentation process, motivated by the data augmentation process of several works [13, 36, 37, 51, 53, 112, 116], we employ `gpt-3.5-turbo-0613` APIs with a temperature of 1.0. Then we specially design a augmentation prompt for RAG as follow:

---

**Query Augmentation Prompt**

You are an AI assistant helping me rewrite the query. I will give you the original query, reference document, title and rewriting requirements. Please rewrite the query based on the following information:

**Original Query**: {Query}
**Reference Documents**: {Top-K Documents}
**Title**: {Title}
**Augmentation Requirements**: {Augmneted Requirements}
**New Queries:**

---

## B.3 Implementation Details

Here, we report our detailed information of DPA-RAG, as a retriever-reranker-reader architecture:

For retriever, following the previous works [12, 52, 65], we utilize Dense Document Retriever (DPR) [30] for encoding documents and questions respectively. After that, we use it retrieves the top 100 relevant Wikipedia documents [95] according to the dot-product similarity.

For reranker, we use the BGE [105] as our backbone model. Specifically, we adjust our batch size to 16. We fine-tune our reranker for 10 epochs and set the learning rate to 1e-5. We utilize the BGE reranker to order the top 100 retrieved documents to obtain the top-3 results.[7].

For the QA fine-tuning setting, we employ the AdamW optimizer [49] to train our LLMs for 3 epochs. Moreover, we set our training batch size to 128. We use eight A100 80g GPUs to fine-tune all models with top-3 documents. Our learning rate is set as 7e-5 with a 3% warmup process. For all experiments, we conduct them using the LLaMA Factory framework [123] with model's default system prompts. We use the version 0.6.3[8] for training LLaMA2, Mistral, Qwen1.5 and Phi2. In addition, we use the version 0.8.1 [9] for Qwen2 and LLaMA3. We report the average performance from five experiments, each with a different random seed.

To facilitate the reproduction of our results, all datasets and evaluation benchmarks used in our experiments have been open-sourced and their detailed sources are indicated. We promise to open-source our code after the blind review process.

## B.4 Baselines

We mainly compare DPA-RAG with multiple strong baselines by using reranker-based methods and preference aligned methods for RAG as follow:

**Reranker-based Baselines:**

- **RankGPT** [90] leverages listwise prompting and utilizes specific distillation method to replicate the document reranking abilities of GPT-3.5 within a smaller ranking model.

---

[7]we use mDeberta as our filtering model, which can be downloaded at https://huggingface.co/MoritzLaurer/mDeBERTa-v3-base-xnli-multilingual-nli-2mil7
[8]https://github.com/hiyouga/LLaMA-Factory/releases/tag/v0.6.3
[9]https://github.com/hiyouga/LLaMA-Factory/releases/tag/v0.8.1

- **LRL** [57] is a model that utilizes GPT-3.5 as a zero-shot reranker for listwise ranking, which directly generates a ranking list of candidate documents.
- **PRP** [74], Pairwise Ranking Prompting, which involves submitting a query alongside a pair of documents into the prompt, enabling large language models to perform ranking tasks.
- **RankLLaMA** [56], based on LLaMA, is trained as a pointwise reranker. This approach involves passing both query and document together to the model. RankLLaMA generates a similarity score reflecting the document's relevance to the query.
- **BGE** [105] is a general Embedding Model developed by BAAI. The reranker use the cross-encoder structure to do full-attention on the input pair.
- **BCEmbedding** [62], Bilingual and Crosslingual Embedding in English and Chinese, developed by NetEase Youdao. Their Reranker is particularly proficient at refining search results and improving ranking tasks.
- **ColBERTv2** [81], a model employs a combination of denoised supervision and residual compression techniques, utilizing token-level decomposition during late interaction.

**Preference-aligned Baselines:**

- **KnowPAT** [120] is a framework that constructs a knowledge-able preference set to align model preferences with knowledge. This framework effectively guides language models to select relevant knowledge for specific inquiries, enhancing their ability to provide pertinent information.
- **REPLUG** [87] It is a retrieval-enhanced language modeling framework that dynamically optimizes the retriever through the output probability of a black box large language model.
- **RA-Judgement** [78], which is known as Retrieval-augmented judgement. In this work, authors explores the knowledge boundary problem of RAG and proposes two experimental settings, Priori Judgment and Posteriori Judgment. RA-judgment is a dynamic improvement method based on Priori Judgment, which can better capture factual information.
- **RRHF** [117] is a training paradigm, which aims to align probabilities of model responses with human preferences by a ranking loss, which can retain the performance of Proximal Policy Optimization (PPO) and is much simpler.
- **RAFT** [119] boosts a language model's proficiency in answering questions within a specific domain by teaching it to disregard irrelevant documents and reference pertinent segments from retrieved texts. It enhances the model's reasoning capabilities and effectiveness in domain-related tasks while maintaining resilience against incorrect retrievals.
- **FILCO** [99] It is a data selection method based on vocabulary and information theory to improve the quality of generated answers provided to the generative model by filtering useful context in the training data.

Furthermore, We also provide a detailed introduction to the **LLM reader model** used by DPA-RAG:

- **LLaMA2** [92] is an upgraded version of LLaMA developed by MetaAI. It utilizes more robust data cleaning and mixing techniques, and up-samples sources closest to factual information,

which can enhance knowledge and reduce hallucinations. Additionally, it employs Grouped-Query Attention technology to lessen reliance on memory.

- **LLaMA3** [59], created by MetaAI, the newest version of the LLaMA series, LLaMA3, includes major enhancements. In contrast to LLaMA2, LLaMA3 incorporates a larger training dataset, extended context length, and an enriched vocabulary, leading to better performance on a range of tasks. Additionally, LLaMA3 offers notable improvements in contextual comprehension and language generation, setting it apart from its predecessor.
- **Qwen1.5** [5] series, created by Alibaba, comprises language models with advanced features like SwiGLU activation, attention QKV bias, group query attention, and a combination of sliding window and full attention mechanisms. These models boast robust fundamental abilities, particularly in language comprehension.
- **Qwen2** [5], developed by Alibaba, is available in several sizes: Qwen2-0.5B /1.5B /7B and 72B. This model is trained on data sources spanning 29 kinds of languages, enabling it to perform exceptionally well in multilingual tasks. Additionally, Qwen2 exhibits strong capabilities in coding and mathematics. Qwen2-72B-Instruct is notable for its ability to handle input windows of up to 128K tokens in length, making it exceptionally well-suited for processing long texts and tackling complex tasks.
- **Mistral** [22], a language model boasting 7 billion parameters, is engineered by Mistral AI for exceptional performance and efficiency. Mistral 7B utilizes Packet Query Attention to accelerate inference and integrates Sliding Window Attention to efficiently manage sequences of varying lengths, all while minimizing inference costs.
  **Phi2** [19], proposed by Microsoft, is a powerful small language model with 2.7 billion parameters. Despite its relatively modest size, Phi-2 demonstrates exceptional reasoning and language comprehension capabilities. At its release, it showcased great performance among small foundational LLMs. In different benchmark tests, model's performance was comparable to, or even surpassed, models that are 25 times larger.
- **GPT-3.5 and GPT-4** [67], proposed by OpenAI, which are part of the GPT families that incorporate a multi-step reinforcement learning from human feedback (RLHF) techniques. the algorithm not only enhances the models' instruction-following ability but also significantly reduces the likelihood of producing harmful or toxic content. Moreover, GPT-4 introduces support for image inputs and attains human-like performance on a range of benchmarks.

## C More Details about Experimental Results

### C.1 Detailed Results for Ablation Studies

Table 6 presents the detailed ablation results of our DPA-RAG across three key phases, with "w/o" indicating the model's version without a particular module. Our findings are as follows:

- DPA-RAG's result declines when any of its components are removed, further validating the necessity of each part we designed.
- Focusing on the Preference Knowledge Construction stage, we notice that the Query Augmentation methods lead to a substantial improvement in performance, which is in line with our expectations. These strategies introduce additional supervision signals

**Table 6: Detailed Ablations of LLaMA2-7B on NQ and TQA. Point-wise., Pair-wise., CPA denotes Point-wise, Pair-wise and Contrastive Preference Alignment respectively.**

| Method | NQ | | TQA | |
|---|---|---|---|---|
| | Hits@1 | F1 | Hits@1 | F1 |
| LLaMA2-7B DPA-RAG | 56.03 | 60.19 | 70.16 | 70.29 |
| *Preference Knowledge Construction* | | | | |
| w/o Query Aug. | -2.13 | -2.31 | -2.62 | -2.87 |
| w/o Filtering. | -0.92 | -0.71 | -1.39 | -1.45 |
| *Multi-Grained Distillation Alignment* | | | | |
| w/o point-wise. | -1.95 | -2.12 | -2.43 | -2.43 |
| w/o pair-wise. | -0.98 | -0.92 | -1.51 | -1.74 |
| w/o CPA | -1.54 | -1.12 | -1.84 | -2.13 |
| w/o MGDA-UB. | -0.52 | -0.77 | -0.84 | -1.10 |
| *Knowledge Self-Alignment* | | | | |
| w/o Pre-Align. | -1.72 | -1.76 | -2.21 | -2.45 |
| LLaMA2-7B RAG | 50.94 | 54.76 | 63.90 | 63.80 |

during the training stages of both the Reranker and the Reader, yielding a joint boost to the DPA-RAG framework. Moreover, the quality filtering process also brings slight performance gains, underscoring the importance of maintaining intent consistency between original and augmented data.

- In the multi-grained distillation alignment stage, each task independently provides stable gains in both NQ and TQA. Point-wise preference alignment, as a fundamental capability for distinguishing knowledge preferences, brings the largest gains in aligning LLMs' preferences. Notably, the MGDA-UB strategy further yields stable gains on top of the joint optimization of three tasks, proving the necessity of introducing multi-task balance optimization.
- The pre-aligned phase also shows steady performance gains, especially evident in TQA. In practice, we find that the potential for internal alignment in TQA is even greater than external, differing from NQ and HQA. Therefore, this insight also highlights the necessity of dual alignment to align datasets from different domains.

## C.2 Details about Diverse Query Augmentations

*Case Study of Augmented Queries.* Table 7 shows some samples which are generated by `gpt-3.5-turbo-0613` APIs in the way of different augmneted requirement, respectively. We can observe that the complexity level of the augmented data showcased in the case is generally consistent with the trend of complexity and diversity scores presented in Table 4.

*Tag Review of Training Data.* In section "Discussion on Query Augmentations", we initially explore how the performance outcome is linked to complexity and diversity within the Natural Questions (NQ) dataset. Following the Instag [50], we also carry out an review of the intent tags within the training dataset. We randomly selected 10,000 samples from the final Supervised Fine-Tuning (SFT) data

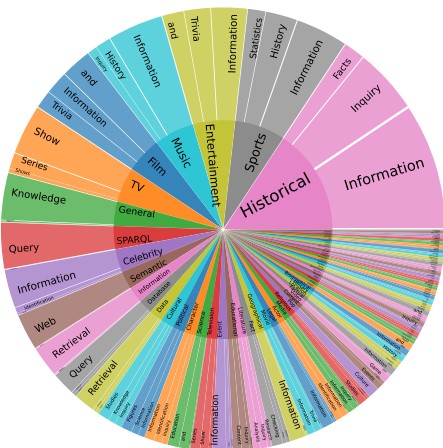

**Figure 7: The sunburst chart displays all tags, with each segment representing the first two words of each tag. The size of each segment is proportional to the tag's frequency.**

pool, which includes both the original data and 5 sets of augmented data. Figure 7 displays the most common tags, which predominantly pertain to historical information, sports-related data, and entertainment queries. The tags are represented by the initial two words, and their size is directly proportional to their frequency. We limit our visualization to only those tags that appear more than 600 times within our dataset.

## C.3 Softer Criterion for Four Conditions

Considering that classifying the four conditions is quite challenging and somewhat absolute, we propose a soft setting as follows:

We calculate the accuracy of direct and document-based answers using the F1 metric, introducing a tunable threshold $x \in (0, 1)$. Answers above this threshold are deemed correct, while those below $1 - x$ are considered incorrect, effectively creating a soft metric ($x = 1$ comparable to the strict setting).

As shown in Table 8, DPA-RAG with the soft metric does yield some improvements, although it does not surpass the results of the original setting. A higher threshold results in fewer preference data, creating a performance trade-off. Our primary motivation for this metric is to better extract preference data from the training set, and we plan to investigate the optimization of the soft metric in our future work.

## C.4 Transfer Experiment of the Reranker

To explore whether there is consistency and transferability among the preferences of different LLMs, we conduct transfer experiments as shown in Table 9. We use Llama2-7B as the reader and test it with aligned GPT-4 and Qwen-2, along with Llama2's own preferred reranker as the transfer reranker. The results indicate that Llama2's own preferences indeed more effectively enhance performance, and surprisingly, other LLMs also exhibit preference transfer effects comparing to standard RAG setting. We believe there may be a certain correlation among the preferences of LLMs, suggesting the possibility of training a universal preference reranker. We will

**Table 7: Examples of different methods for generating new queries.**

| Method | Requirement | Query |
|---|---|---|
| Origin | - | What screenwriter with credits for "Evolution" co-wrote a film starring Nicolas Cage and Téa Leoni? |
| Rephrasing | Rephrase the original query with the same intention. | Who is the screenwriter credited for "Evolution" who also co-authored a movie featuring Nicolas Cage and Téa Leoni? |
| Decomposition | Decompose the original query into several sub-problems. | Sub-problem 1: Identify the screenwriter who has credits for the film "Evolution". Sub-problem 2: Determine if the screenwriter from sub-problem 1 has also co-written a film where Nicolas Cage and Téa Leoni were cast. |
| SPARQL | Rewrite the original query based on the SPARQL syntax and generate it directly. | `SELECT ?screenwriter WHERE {`
`?film rdf:type dbo:Film .`
`?film dbo:writer ?screenwriter .`
`?film dbo:starring dbr:Nicolas_Cage .`
`?film dbo:starring dbr:Tea_Leoni .`
`?screenwriter dbo:film dbr:Evolution .`
`?screenwriter rdfs:label ``David Weissman'' .`
`}` |
| Constraint | Add more conditional and constrained statements to the original query. | Which screenwriter, known for working on the movie "Evolution", also co-authored a screenplay for a feature film that includes Nicolas Cage and Téa Leoni in the cast, and has a history of collaboration with David Diamond? |
| Complexity | Increase the semantic complexity of the original query. | Which scriptwriter, known for his partnership with David Diamond and shared film credits on "Evolution", also co-authored a screenplay that featured Nicolas Cage and Téa Leoni in leading roles, after initially meeting his writing colleague at Akiba Hebrew Academy and making their screenwriting sale debut with "The Whiz Kid" to 20th Century Fox? |

| Strategy | NQ | TQ | HQ |
|---|---|---|---|
| Standard RAG (GPT-3.5) | 48.0 | 74.1 | 32.8 |
| +Soft Strategies (50%) | 48.8 | 74.3 | 28.1 |
| +Soft Strategies (80%) | 50.4 | 75.9 | 33.2 |
| Original Strategies (100%) | 52.8 | 77.1 | 36.1 |

**Table 8: The ablation study on soft metrics (F1).**

| Model | NQ | TQ | HQ | WebQ |
|---|---|---|---|---|
| Standard RAG (Llama2-7B) | 54.8 | 63.8 | 38.9 | 64.2 |
| +GPT4 Reranker | 58.5 | 67.2 | 40.1 | 68.2 |
| +Qwen2 Reranker | 56.3 | 64.7 | 39.3 | 65.5 |
| DPA-RAG (Llama2-7B) | 60.2 | 70.3 | 43.3 | 71.8 |

**Table 9: The transfer experiments for testing the consistency of preference information between different LLMs**

focus on the consistency of preferences as a direction for future exploration.

## C.5 Generalization Experiments of Different Retrievers

To verify that the DPA-RAG method can be adapted to different types of retrievers, we supplement more results for Sparse (**BM25**) and Dense Retriever (**RocketQA v2** [75], **Contriever** [18]) in Table 10. DPA-RAG still demonstrates significant results, even with the superior performance of Contriever. This aligns with our motivation; although the retriever's vector similarity is high, it doesn't fully

align with the preferences of LLMs. Thus, DPA-RAG can further mitigate unaligned knowledge.

## C.6 Comparison Experiments with Noise Robustness Methods

**Definition Clarification.**

As two easily confused topics, we first want to clarify that preference alignment and noise robustness methods for RAG are significantly different in definition, to help reviewers and readers distinguish between these two topics.

| Retriever | Reader | NQ | TQ | HQ |
|---|---|---|---|---|
| BM25 | Qwen1.5 | 42.7 | 46.3 | 40.3 |
| | +DPA-RAG | 46.3 (+3.6) | 50.3 (+4.0) | 42.8 (+2.5) |
| | Llama3 | 46.8 | 52.5 | 41.2 |
| | +DPA-RAG | 50.1 (+3.3) | 56.9 (+4.4) | 43.7 (+2.5) |
| RocketQAv2 | Qwen1.5 | 52.76 | 47.35 | 38.63 |
| | +DPA-RAG | 53.5 (+0.74) | 54.2 (+6.85) | 39.3 (+0.67) |
| | Llama3 | 56.1 | 54.7 | 39.6 |
| | +DPA-RAG | 58.2 (+2.1) | 58.8 (+4.1) | 41.3 (+1.7) |
| Contriever | Qwen1.5 | 54.3 | 69.9 | 39.8 |
| | +DPA-RAG | 56.1 (+1.8) | 71.1 (+1.2) | 42.4 (+2.6) |
| | Llama3 | 58.5 | 71.0 | 40.4 |
| | +DPA-RAG | 61.9 (+3.4) | 73.9 (+2.9) | 43.4 (+3.0) |

**Table 10: The results of DPA-RAG with other sparse and dense retrievers (F1).**

**Noise Robustness:** these studies [110, 111, 113] aim to filtering noise from the perspective of users or queries, such as the counterfactual information [10]. The goal of denoising is to retain Docs containing grounding / relevant knowledge while excluding disruptive information.

**Preference Alignment (PA):** Unlike them, the PA in our paper considers LLM's aspect and highlights that highly relevant retrieved knowledge do not exactly match the knowledge required for LLM reasoning (see Figure 1). Therefore, the goal of the PA is not to retain documents containing grounding / relevant information, but to preserve Docs consistent with the model's preferences. This implies that **while highly relevant knowledge is not objectively noise, it may still be unaligned knowledge**.

**Results.** To ensure comprehensiveness, we further compare our approach with three widely used noise robustness methods for RAG systems. Additionally, we maintain the consistency of the retriever and reader models to facilitate a fair comparison.

- **RankRAG** [114]: Unifying RAG and ranking with instruction tuning.
- **EOR** [39]: a trainable framework that can adaptively retrieve from different knowledge sources and effectively decrease unpredictable reader errors.
- **InstructRAG** [102]: Let LMs explicitly learn the denoising process through self-synthesized rationales

The results show that despite these advanced denoising methods, DPA-RAG still demonstrates further significant improvements. This not only illustrates the effectiveness of our method but also emphasizes that preference alignment remains an inherent bias among RAG system components, which cannot be resolved by simple denoising.

## C.7 Comparison Experiments with Other RAG Baselines

To validate the exceptional effectiveness of DPA-RAG, we further compared it with two classic RAG baseline methods: Self-RAG [2] and LLM Rewriting.

| Model | NQ | TQ | HQ |
|---|---|---|---|
| RankRAG (Llama3-8b) | 50.6 | 82.9 | 35.3 |
| DPA-RAG (Llama3-8b) | 57.4 | 82.9 | 37.9 |
| EOR (Llama2-7B-chat) | 50.2 | 75.9 | - |
| DPA-RAG (Llama2-7B-chat) | 56.8 | 77.1 | - |
| Instruct RAG (Llama3-8B-instruct) | 64.2 | 76.8 | - |
| DPA-RAG (Llama3-8B-instruct) | 66.1 | 78.3 | - |

**Table 11: The comparison (EM) between DPA-RAG and noise robust RAG baselines. To ensure a fair comparison, we adopt the same settings used in the papers of these baselines for our experiments (e.g. RankRAG uses Top5 Docs).**

| Methods | Reader | NQ | TQ | HQ |
|---|---|---|---|---|
| Self-RAG (GPT 3.5) | GPT 3.5 | 48.9 | 64.2 | - |
| LLM Rewrite (GPT 3.5) | GPT 3.5 | 39.4 | 46.3 | 26.4 |
| DPA-RAG (GPT 3.5) | GPT 3.5 | 52.8 | 77.1 | 36.1 |
| LLM Rewrite (GPT 4) | GPT 4 | 41.4 | 46.3 | 32.7 |
| DPA-RAG (GPT 4) | GPT 4 | 53.3 | 80.1 | 37.7 |

**Table 12: Results on SelfRAG and LLM Rewriting of GPT series (F1).**

First, we do not include Self-RAG in the main text because its design focuses on determining when to retrieve information to reduce hallucinations, while DPA-RAG concentrates on aligning the retrieved knowledge. Additionally, Self-RAG requires multiple rounds of retrieval and responses, whereas DPA-RAG operates with a single retrieval. Nonetheless, we follow the FlashRAG [25] framework to replicate Self-RAG. As shown in Table 7, the results indicate that DPA-RAG has a clear performance advantage, particularly in the TQ task, where the F1 score improves by over 10%.

Next, we evaluated the LLM Rewriting strategy using prompt templates [55] from previous research. Results in Table 4 show that whether using GPT-3.5 or GPT-4 as the reader, DPA-RAG consistently demonstrates stable gains, exceeding 5%.

In summary, by comparing these two RAG baseline strategies, we further confirmed the generalization ability of DPA-RAG.

## C.8 The Performance on Instruct Version

In the main text, we use the base version and the same SFT QA data throughout the paper to ensure fairness. We also supplement the results of Llama3-8B Instruct in table 13. The results show that the Instruct version still has some performance gaps compared to SFT on specific datasets (NQ, HQ) and DPA-RAG continues to show stable improvements in this setup.

| Model | NQ | TQ | HQ | WebQ |
|---|---|---|---|---|
| Llama3-8b Instruct | 57.9 | 71.8 | 35.3 | 81.9 |
| +DPA RAG | 59.5 | 73.2 | 37.1 | 82.4 |
| Qwen2-7b Instruct | 53.1 | 71.6 | 29.6 | 77.5 |
| +DPA RAG | 55.6 | 73.4 | 30.9 | 78.7 |

**Table 13: The results of DPA-RAG with instruct version models (F1).**

## C.9 Case Studies for Preference Alignment

To gain a deeper understanding of how DPA-RAG aligns the preferences of large models to knowledge, we conducted a case study and manual analysis, marking key supporting knowledge and misleading knowledge in the TOP-3 documents with green and red, respectively.

---

**NQ-Case1**

### Baseline

**Query**: Where was the diary of a wimpy kid filmed ?

**Reference Documents**:
**document1:**
Title: Diary of a Wimpy Kid: Rodrick Rules (film)
Content: The film was released on March 25, 2011. Talks of a sequel were announced after the release of the first, but was not officially announced until May 12, 2010, announcing that it would be released March 25, 2011. Filming took place in Vancouver, British Columbia and New Westminster, British Columbia from August 23 to October 27, 2010. The mall scene was filmed at Park Royal Mall in West Vancouver. Director Thor Freudenthal was replaced by director David Bowers ('Flushed Away" and "Astro Boy"). The film's trailer was shown with "Gulliver's Travels" on December 25, 2010. It was later online on
**document2:**
Title: Diary of a Wimpy Kid: Dog Days (film)
Content: The outdoor pool scenes at the beginning and the end of the movie were filmed at Steveston Outdoor pool in Richmond, BC. Filming occurred at the Steveston Outdoor pool during the beginning of September 2011. The Chinese Bunkhouse at the Steveston Shipyards in Richmond, BC was the location of the Wilderness Explorers cabin for Troop 133. During filming, stars Zachary Gordon and Robert Capron were seen riding the Corkscrew at Playland at Vancouver's fair, the PNE. A poster was leaked in March 2012. A teaser trailer was attached to "The Three Stooges". An advance screening for the film was held on
**document3:**
Title: Diary of a Wimpy Kid (film series)
Content: The "Diary of a Wimpy Kid" film series was released on March 25, 2011 and is based on the second book, "Rodrick Rules" with scenes from "The Last Straw". Principal photography began on August 23, 2010, and was completed on October 27, 2010, with filming taking place in Vancouver and New Westminster. "Rodrick Rules" was directed by David Bowers, with Zachary Gordon reprising his role as Greg Heffley. New main characters include Holly Hills (Peyton List), Grandpa (Terence Kelly), and Bill Walter (Fran Kranz). Edward Shearmur composed the original score for the film. "Diary of a Wimpy Kid: Dog

---

Days" is the third film in the series.

**Output**: New Westminster, British Columbia ✗

\*\*\*\*\*\*\*\*\*\*\*\*\*\*\*\*\*\*\*\*\*\*\*\*\*\*\*\*\*\*\*\*\*\*\*\*\*\*\*\*\*\*\*\*\*\*\*\*\*\*\*\*\*\*\*\*\*\*\*\*\*\*\*\*

### DPA-RAG

**Query**: Where was the diary of a wimpy kid filmed ?

**Reference Documents**:
**document1:**
Title: Diary of a Wimpy Kid: Dog Days (film)
Content: Outdoor pool scenes at the beginning and the end of the movie were filmed at Steveston Outdoor pool in Richmond, B.C. Filming occurred at the Steveston Outdoor pool during the beginning of September 2011. The Chinese Bunkhouse at the Steveston Shipyards in Richmond, B.C. was the location of the Wilderness Explorers cabin for Troop 133. During filming, stars Zachary Gordon and Robert Capron were seen riding the Corkscrew at Playland at Vancouver's fair, the P.N.E. A poster was leaked in March 2012. A teaser trailer was attached to "The Three Stooges". An advance screening for the film was held on...

**document1:**
Title: Diary of a Wimpy Kid: Dog Days (film)
Content: All hang out together at the local pool. Greg describes that his summer did not go the way he expected, but had positive circumstances. This film is the last movie in the "Diary of a Wimpy Kid" film series to feature the original cast, as they aged out of their roles as middle-schoolers. Principal photography began on August 8, 2011, in Vancouver and was completed on October 7, 2011. The location for the country club pool was Eagle Ridge Outdoor pool in Coquitlam, B.C. Filming at Eagle Ridge Outdoor pool took place during the end of August 2011. The municipal...

Title: Diary of a Wimpy Kid (film series)
Content: "Diary of a Wimpy Kid" film series. It was released on March 25, 2011, and is based on the second book, "Rodrick Rules" with scenes from "The Last Straw". Principal photography began on August 23, 2010, and was completed on October 27, 2010, with filming taking place in Vancouver and New Westminster. "Rodrick Rules" was directed by David Bowers, with Zachary Gordon reprising his role as Greg Heffley. New main characters include Holly Hills (Peyton List), Grandpa (Terence Kelly), and Bill Walter (Fran Kranz). Edward Shearmur composes the original score for the film. "Diary of a Wimpy Kid: Dog Days" is the third film...

**Output**: Vancouver ✓

\*\*\*\*\*\*\*\*\*\*\*\*\*\*\*\*\*\*\*\*\*\*\*\*\*\*\*\*\*\*\*\*\*\*\*\*\*\*\*\*\*\*\*\*\*\*\*\*\*\*\*\*\*\*\*\*\*\*\*\*\*\*\*\*

**Analysis**: The retrieved documents of the baseline contain both aligned knowledge and unaligned knowledge, with the final reasoning being misled by the unaligned knowledge. DPA-RAG filters out the unaligned knowledge during the Reranker process, retaining only the aligned knowledge, leading to successful reasoning in the end.

## NQ-Case2

### Baseline

**Query**: Three largest cities in the world by population?

**Reference Documents**:
**document1:**
Title: Americas
Content: Three public bodies of the Netherlands, two unincorporated territories of the United States, and one uninhabited territory of the United States. In 2015 the total population of the Americas was about 985 million people, divided as follows: There are three urban centers that each hold titles for being the largest population area based on the three main demographic concepts: In accordance with these definitions, the three largest population centers in the Americas are: Mexico City, anchor to the largest metropolitan area in the Americas; New York City, anchor to the largest urban area in the Americas; and São Paulo, the...

**document2:**
Title: Europe
Content: Are recognised political goals in Europe today. The Council of Europe Framework Convention for the Protection of National Minorities and the Council of Europe's European Charter for Regional or Minority Languages set up a legal framework for language rights in Europe. The four most populous cities of Europe are Istanbul, Moscow, Paris and London, each have over 10 million residents, and as such have been described as megacities. While Istanbul has the highest total population, one third lies on the Asian side of the Bosporus, making Moscow the most populous city entirely in Europe. The next largest cities in order...

**document3:**
Title: World population

Content: Permanently inhabited on a large scale. Asia is the most populous continent, with its 4.54 billion inhabitants accounting for 60% of the world population. The world's two most populated countries, China and India, together constitute about 36% of the world's population. Africa is the second most populated continent, with around 1.28 billion people, or 16% of the world's population. Europe's 742 million people make up 10% of the world's population as of 2018, while the Latin American and Caribbean regions are home to around 651 million (9%). Northern America, primarily consisting of the United States and Canada, has a population...

**Output**: Istanbul ✗

---

```
************************************************************
```

### DPA-RAG

**Query**: Three largest cities in the world by population?

**Reference Documents**:
**document1:**
Title: Beijing
Content: Resided in urban districts or suburban townships, and 2.897 million lived in rural villages. The encompassing metropolitan area was estimated by the OECD (Organisation for Economic Co-operation and Development) to have, a population of 24.9 million. Within China, the city ranked second in urban population after Shanghai and the third in municipal population after Shanghai and Chongqing. Beijing also ranks among the most populous cities in the world, a distinction the city has held for much of the past 800 years, especially during the 15th to early 19th centuries when it was the largest city in the world. About...

**document2:**
Title: City
Content: A fifth of the population is said to live in shantytowns (favelas, poblaciones callampas, etc.). Batam, Indonesia, Mogadishu, Somalia, Xiamen, China, and Niamey, Niger, are considered among the world's fastest-growing cities, with annual growth rates of 5–8%. In general, the more developed countries of the "Global North" remain more urbanized than the less developed countries of the "Global South"—but the difference continues to shrink because urbanization is happening faster in the latter group. Asia is home to by far the greatest absolute number of city-dwellers: over two billion and counting. The UN predicts an additional 2.5 billion city dwellers ...

**document3:**
Title: Russia
Content: Russia, officially the Russian Federation, is a country in Eurasia. At, Russia is the largest country in the world by area, covering more than one-eighth of the Earth's inhabited land area, and the ninth most populous, with about 144.5 million people, excluding Crimea. About 77% of the population live in the western, European part of the country. Russia's capital, Moscow, is the largest metropolitan area in Europe proper and one of the largest cities in the world; other major cities include Saint Petersburg, Novosibirsk, Yekaterinburg, and Nizhny Novgorod. Extending across the entirety of Northern Asia...

**Output**: Shanghai ✓

**************************************************************

**Analysis**: The retrieved documents of the baseline only contain unaligned knowledge, with the final reasoning being misled by the unaligned knowledge. DPA-RAG filters out the unaligned knowledge during the reranking process. Moreover, it also find the aligned knowledge from documents with lower vector similarity, reranking them to the front. Finally, the aligned knowledge leads to successful reasoning in the end.

## TQA-Case1

### Baseline

**Query**: Which volcano in Tanzania is the highest mountain in Africa?

**Reference Documents**:
**document1:**
Title: Mount Kilimanjaro
Content: Mount Kilimanjaro or just Kilimanjaro, with its three volcanic cones, "Kibo," "Mawenzi," and "Shira," is a dormant volcano in Tanzania. It is the highest mountain in Africa, about [height missing] from its base, and [elevation missing] above sea level. The first people known to have reached the summit of the mountain were Hans Meyer and Ludwig Purtscheller in 1889. The mountain is part of the Kilimanjaro National Park and is a major climbing destination. The mountain has been the subject of many scientific studies because of its shrinking glaciers and disappearing ice fields. Kilimanjaro is a large stratovolcano and is composed...

**document2:**
Title: Mount Kilimanjaro
Content: Of three distinct volcanic cones: Kibo, the highest; Mawenzi at; and Shira, the shortest at. Mawenzi and Shira are extinct, while Kibo is dormant and could erupt again. Uhuru Peak is the highest summit on Kibo's crater rim. The Tanzania National Parks Authority, a Tanzanian governmental agency, and the United Nations Educational, Scientific and Cultural Organization list the height of Uhuru Peak as [height missing]. That height is based on a British Ordnance Survey in 1952. Since then, the height has been measured as [height missing] in 1999, [height missing] in 2008, and [height missing] in 2014. The interior of the volcanic edifice is poorly...

**document3:**
Title: Mount Kilimanjaro
Content: Mount Kilimanjaro or just Kilimanjaro, with its three volcanic cones, "Kibo," "Mawenzi," and "Shira," is a dormant volcano in Tanzania. It is the highest mountain in Africa, about ; from its base, and above sea level. The first people known to have reached the summit of the mountain were Hans Meyer and Ludwig Purtscheller in 1889. The mountain is part of...

**Output**: Mawenzi ✗

************************************************************

### DPA-RAG

**Query**: Which volcano in Tanzania is the highest mountain in Africa?

**Reference Documents**:
**document1:**
Title: Mount Kilimanjaro
Content: Mount Kilimanjaro or just Kilimanjaro, with its three volcanic cones, "Kibo," "Mawenzi," and "Shira," is a dormant volcano in Tanzania. It is the highest mountain in Africa, about ; from its base, and above sea level. The first people known to have reached the summit of the mountain were Hans Meyer and Ludwig Purtscheller in 1889. The mountain is part of...

**document2:**
Title: Mount Kilimanjaro
Content: Mount Kilimanjaro or just Kilimanjaro, with its three volcanic cones, "Kibo," "Mawenzi," and "Shira," is a dormant volcano in Tanzania. It is the highest mountain in Africa, [height missing] from its base, and [altitude missing] above sea level. The first people known to have reached the summit of the mountain were Hans Meyer and Ludwig Purtscheller in 1889. The mountain is part of the Kilimanjaro National Park and is a major climbing destination. The mountain has been the subject of many scientific studies because of its shrinking glaciers and disappearing ice fields. Kilimanjaro is a large stratovolcano and is composed...

**document3:**
Title: Geography of Tanzania
Content: Geography of Tanzania Tanzania comprises many lakes, national parks, and Africa's highest point, Mount Kilimanjaro [altitude missing]. Northeast Tanzania is mountainous and includes Mount Meru, an active volcano, Mount Kilimanjaro, a dormant volcano, and the Usambara and Pare mountain ranges. Kilimanjaro attracts thousands of tourists each year. West of those mountains is the Gregory Rift, which is the eastern arm of the Great Rift Valley. On the floor of the rift are a number of large salt lakes, including Natron in the north, Manyara in the south, and Eyasi in the southwest. The rift also encompasses the Crater Highlands, which includes...

**Output**: Kilimanjaro ✓
************************************************************

**Analysis**: The baseline's retrieved documents include both aligned and unaligned knowledge, with unaligned knowledge ranked higher due to its similarity, leading to incorrect inferences. Conversely, DPA-RAG sorted aligned knowledge to top-1 and filtered out unaligned knowledge in reranking process, ultimately succeeding in inference.

**TQA-Case2**

## Baseline

**Query**: What nationality was the painter Vincent van Gogh?

**Reference Documents**:
**document1:**
Title: Vincent van Gogh
Content: Vincent Willem van Gogh (30 March 1853–29 July 1890) was a Dutch Post-Impressionist painter who is among the most famous and influential figures in the history of Western art. In just over a decade, he created about 2,100 artworks, including around 860 oil paintings, most of them in the last two years of his life. They include landscapes, still lifes, portraits, and self-portraits, and are characterized by bold colors and dramatic, impulsive, and expressive brushwork that contributed to the foundations of modern art. However, he was not commercially success-ful, and his suicide at 37 followed years...
**document2:**
Title: Theo van Gogh (art dealer)
Content: Theodorus "Theo" van Gogh was born on 1 May 1857 in the village Groot-Zundert in the province of North Brabant, Netherlands. He was the son of Theodorus van Gogh and Anna Cornelia Carbentus. His elder brother was artist Vincent van Gogh (1853–1890). Theo worked for some years at the Dutch office of the Parisian art dealers Goupil & Cie in The Hague. Theo joined the Brussels office on 1 January 1873 as their youngest employee. After Theo was transferred to the London office, he moved to the office in...

**document3:**
Title: Vincent van Gogh
Content: Van Gogh Museum opened in the Museumplein in Amsterdam in 1973. It became the second most popular museum in the Netherlands, after the Rijksmuseum, regularly receiving more than 1.5 million visitors a year. In 2015 it had a record 1.9 million; 85 percent of the visitors come from other countries. Vincent Willem van Gogh (30 March 1853–29 July 1890) was a Dutch Post-Impressionist painter who is among the most famous and influential figures in the history of Western art. In just over a decade, he created about 2,100 artworks, including around 860 oil paintings, most of...

**Output**: Autochtones ✗

**************************************************************

## DPA-RAG

**Query**: Three largest cities in the world by population?

**Reference Documents**:
**document1:**
Title: Vincent van Gogh
Content: The Van Gogh Museum opened in the Muse-umplein in Amsterdam in 1973. It became the second most popular museum in the Netherlands, after the Rijksmuseum, regularly receiving more than 1.5 million visitors a year. In 2015, it had a record 1.9 million; 85 percent of the visitors come from other countries. Vincent Willem van Gogh (30 March 1853 – 29 July 1890) was a Dutch Post-Impressionist painter who is among the most famous and influential figures in the history of Western art. In just over a decade, he created about 2,100 artworks, including around 860 oil paintings, most of...

**document2:**
Title: Vincent van Gogh
Content: Vincent Willem van Gogh (30 March 1853 – 29 July 1890) was a Dutch Post-Impressionist painter who is among the most famous and influential figures in the history of Western art. In just over a decade, he created about 2,100 artworks, including around 860 oil paintings, most of them in the last two years of his life. They include landscapes, still lifes, portraits, and self-portraits, and are characterised by bold colours and dramatic, impulsive and expressive brushwork that contributed to the foundations of modern art. However, he was not commercially successful, and his suicide at 37 followed years...

**document3:**
Title: Vincent van Gogh
Content: Vincent Willem van Gogh was born on 30 March 1853 into a Dutch Reformed family in Groot-Zundert, in the predominantly Catholic province of North Brabant in the southern Netherlands. He was the oldest surviving child of Theodorus van Gogh, a minister of the Dutch Reformed Church, and Anna Cornelia Carbentus. Van Gogh was given the name of his grandfather, and of a brother stillborn exactly a year before his birth. Vincent was a common name in the Van Gogh family: his grandfather, Vincent (1789 – 1874), who received a degree in theology at the University of Leiden in 1811, had six...

**Output**: Dtuch ✓

**************************************************************

**Analysis**: Although the baseline's retrieved documents only contains aligned knowledge, the quantity and order of relevant knowledge are relatively low. DPA-RAG not

only sorts multiple aligned pieces of knowledge to the front during the reranking process, but also relies on the key information capture ability brought by the pre-aligned stage, allowing the LLMs to better focus on knowledge beneficial to inference, ultimately leading to successful reasoning.

Received 20 February 2007; revised 12 March 2009; accepted 5 June 2009

