# OpenReview forum: "Understand What LLM Needs: Dual Preference Alignment for Retrieval-Augmented Generation"
_ACM.org/TheWebConf/2025/Conference — WWW 2025 Poster_

### Official Review · Reviewer_hkAH · 2024-11-30

**Novelty:** 5
**Technical Quality:** 6

**Review:**

This paper presents a novel framework called DPA-RAG that aims to address the preference gap between the retriever and the language model (LM) reader in retrieval-augmented generation (RAG) systems. The key contributions include:

Clearly defining the problem - the retrieved documents may not contain the knowledge required by the LM for reasoning, leading to a preference gap between the retriever and the LM, which can negatively impact the performance of RAG systems.
Proposing the DPA-RAG framework, with three main components: preference knowledge construction, reranker-LLM alignment, and LLM self-alignment, which effectively mitigate the above problem.
Conducting extensive experimental evaluation on four knowledge-intensive QA datasets, with results showing DPA-RAG outperforming various strong baselines.

Overall, this is an excellent work that presents a valuable solution to an important problem in retrieval-augmented generation. The well-designed framework and comprehensive experimental evaluation represent a significant contribution to the field. The paper could further explore the method's limitations and potential future directions.

**Questions:**

* Could the authors provide more insights into the potential limitations of their approach, particularly in terms of the scalability and generalizability to different types of tasks or domains?
* How sensitive is the performance of DPA-RAG to the choice of the reranker architecture and the specific fine-tuning hyperparameters? Are there any guidelines or principles the authors can share for selecting these components?
* The authors mention the potential for training a universal preference reranker based on the observed transfer effects between different LMs. Could the authors elaborate on this idea and discuss any potential challenges or future research directions in this area?
* The authors indicate that the pre-alignment stage is crucial for the LM's internal self-alignment. Could the authors provide more details on the rationale and intuition behind this stage, and discuss any potential alternative approaches that could achieve similar effects?
* In the case studies, the authors demonstrate the ability of DPA-RAG to filter out unaligned knowledge. Could the authors comment on the potential risks or downsides of this filtering process, and how they ensure that relevant knowledge is not inadvertently removed?

**Reviewer Confidence:**

4: The reviewer is certain that the evaluation is correct and very familiar with the relevant literature

**Scope:**

4: The work is relevant to the Web and to the track, and is of broad interest to the community

---

### Official Review · Reviewer_beCQ · 2024-11-30

**Novelty:** 5
**Technical Quality:** 5

**Review:**

This work presents DPA-RAG, a novel approach for enhancing Retrieval-Augmented Generation (RAG) systems by focusing on preference alignment. The authors address the issue of misalignment between retrieved documents and the specific knowledge needs of Large Language Models (LLMs).

Quality:
1. The authors meticulously design a three-step process for constructing preference knowledge. They begin by extracting samples from the training data that reflect LLM knowledge preferences, labeling them as "Aligned Knowledge" or "Unaligned Knowledge". This crucial step ensures the collected data aligns with the specific knowledge needs of the LLM.
2. Five diverse query augmentation strategies are employed to address data scarcity and enhance the quality of preference knowledge. These strategies introduce various ways to rephrase or enrich the original query, thereby increasing data complexity and diversity, leading to improved model performance.
3. The use of a Natural Language Inference (NLI) model as a quality filtering mechanism ensures that augmented queries maintain semantic consistency with the original queries, further enhancing data quality.

Clarity:
The paper is well-written, with a clear structure and concise explanations. The authors effectively use figures and tables to illustrate their methodology and present their results. The overall framework of DPA-RAG, including preference knowledge construction, dual preference alignment, and the inference process, is clearly depicted.

Originality:
1. Dual Preference Alignment: The core contribution lies in the dual preference alignment strategy, which aims to align both the reranker (external alignment) and the LLM (internal alignment) with the LLM's knowledge preferences:
	(1). External Alignment: Achieved by fine-tuning the reranker with multi-grained alignment tasks, including point-wise, pair-wise, and contrastive preference alignment. This ensures the reranker prioritizes documents containing knowledge preferred by the LLM.
	(2). Internal Alignment: A pre-aligned stage is introduced before the Supervised Fine-tuning (SFT) stage, allowing the LLM to implicitly capture knowledge aligned with its reasoning preferences. This process enhances the LLM’s ability to identify and utilize aligned knowledge during inference.
2. Query Augmentation Strategies: The authors propose five unique query augmentation strategies specifically designed for RAG tasks. These strategies aim to increase the complexity and diversity of the preference data, ultimately leading to better alignment and improved model performance.

Significance: The work addresses a critical challenge in RAG systems: aligning retrieved knowledge with the intricate reasoning preferences of LLMs. The proposed dual preference alignment strategy effectively tackles this issue, potentially leading to more reliable and accurate RAG systems.


Pros:
1. Addresses the critical issue of knowledge misalignment in RAG systems.
2. Novel dual preference alignment strategy to align both the reranker and LLM.
3. Innovative query augmentation techniques to improve preference knowledge quality.
4. Comprehensive experiments on four diverse QA datasets demonstrating the effectiveness of DPA-RAG.
5. Detailed analysis and insights for building reliable RAG systems.

Cons:
1. The computational cost of the preference alignment process, particularly the multi-grained distillation alignment, may be significant.
2. The reliance on LLM-generated preference scores for pair-wise preference alignment might introduce bias.
3. The effectiveness of DPA-RAG may vary across different LLMs and domains, requiring further investigation.

**Questions:**

Computational Cost and Efficiency: The authors mention that the computational cost of the preference alignment process, especially the multi-grained distillation alignment, might be significant.
1. Could the authors elaborate on the computational complexity of DPA-RAG compared to other RAG methods?
2. Have the authors considered strategies for optimizing the efficiency of the preference alignment process, especially for large-scale datasets or computationally constrained environments?

Potential Bias in Pair-wise Preference Alignment: The authors acknowledge the potential for bias arising from the reliance on LLM-generated preference scores for pair-wise preference alignment.
1. What steps were taken to mitigate this bias?
2. Are there alternative approaches to scoring documents for pair-wise preference alignment that could reduce or eliminate this bias? For instance, could human feedback be incorporated to provide a more objective assessment of document preference?

**Reviewer Confidence:**

3: The reviewer is confident but not certain that the evaluation is correct

**Scope:**

3: The work is somewhat relevant to the Web and to the track, and is of narrow interest to a sub-community

---

### Official Review · Reviewer_U8a3 · 2024-12-01

**Novelty:** 5
**Technical Quality:** 5

**Review:**

The paper proposes a retrieval-augmented framework to mitigate the problem of hallucination in large language models (LLMs). The framework first learns to identify and rank a set of relevant documents that align with LLMs on QA tasks. It is then tuned to focus on the aligned document within a list of documents while utilizing that document to generate the correct answer to the query.

Pros:
1. The paper proposes a technically sound framework to address the challenging hallucination issue when applying LLMs to QA tasks.
2. The experimental section includes numerous baselines and demonstrates that their method can achieve better performance.
3. Their observation that some relevant documents might harm LLM performance in QA tasks, while some irrelevant documents might help, is quite interesting. They successfully designed a framework based on this observation, which improves the performance of LLMs in QA tasks.

Cons:
1. The authors missed some citations.
2. Some concepts are unclear. For example, "the top-K docs" sometimes appears to refer to the top-K documents retrieved by the query, while other times it seems to refer to documents selected randomly.
3. More details about how they prepared the training and testing datasets are needed for clarification. It seems their framework involves multiple ML tasks. Additionally, it does not appear that their framework can introduce new knowledge. Therefore, I am curious about:
   a). Whether there was any data leakage between the training and testing datasets.
   b). How well the framework performs on unseen data points.

**Questions:**

1. The phrase **"filter out"** in the first paragraph of Section 3 seems unclear. Are the authors referring to removing aligned documents or using only aligned documents in the downstream process?
2. Could the authors provide a clearer definition of **"alignment"** in the paper? What qualifies a document as being aligned with the LLM's knowledge?
3. The term **"hierarchically sample"** in Section 3.2 is unclear. Additionally, why did the authors select four documents? They appear to choose documents from ranking positions 1, 25, 50, and 100, but what is the rationale behind this choice?
4. In their QA task setting, can one query contain two correct answers?
5. The mention of **"SPARQL"** in Section 3.2 requires more explanation and a proper citation.
6. The NLI model referenced in Section 3.2 needs a citation, and its performance data should be clarified.
7. In Figure 2, there are three instances of **"Top-K Docs."** Are these the same document set, or are they actually different sets?
8. The multi-task optimization component is not clearly represented in Figure 2.
9. It is somewhat unclear why the authors require **k documents** in the final step (answer generation). Why can't they simply use the most aligned document as the reference source?
10. Significance tests are missing from Tables 1 through 4.

**Reviewer Confidence:**

3: The reviewer is confident but not certain that the evaluation is correct

**Scope:**

4: The work is relevant to the Web and to the track, and is of broad interest to the community

---

### Official Review · Reviewer_QQsJ · 2024-12-02

**Novelty:** 5
**Technical Quality:** 4

**Review:**

This paper presents a framework DPA-RAG to align diverse knowledge preferences within RAG systems. DPA-RAG consists of reranker-LLM alignment and LLM self-alignment. Experiments on four datasets demonstrate the effectiveness of DPA-RAG.

Strong points:

S1. The paper is well motivated, and the observations in the introduction are helpful for understanding the design principles of RAG systems.

S2. Reranker alignment and LLM self-alignment are well designed to align knowledge form different aspects.

S3. The experiments are extensive.

Weak points:

W1. The data construction process is not clear. See Q1 below.

W2. The cost of DPA-RAG is high. For each dataset, DPA-RAG needs to construct preference data, and train the reranker and LLM, which means that it cannot generalize to different datasets.

W3. DPA-RAG relies on the training set to extract preference knowledge, which often not available in real applications.

**Questions:**

Q1. In preference knowledge extraction, what does "hierarchically sample" mean? What does the notation {i=1, 25, 50, 100} refer to?

Q2. How many preference data is enough? In other words, how do you determine when to stop the augmentation process?

Q3. DPA-RAG relies on the training set D_train, which is hard to acquire in real applications. Please discuss this.

Q4. It seems that the reranker is dataset-specific. For each dataset, it needs to be retrained. However, in real applications, the resource is dynamic and it is unrealistic to train a reranker for each corpus. Please elaborate on this.

Q5. For closed-source LLM, how does self-alignment work? If this component is just for open-source LLM, please clarify it in the paper.

**Reviewer Confidence:**

4: The reviewer is certain that the evaluation is correct and very familiar with the relevant literature

**Scope:**

4: The work is relevant to the Web and to the track, and is of broad interest to the community

---

### Official Review · Reviewer_fGTa · 2024-12-02

**Novelty:** 3
**Technical Quality:** 5

**Review:**

This paper introduces DPA-RAG, a universal framework for aligning diverse knowledge preferences within RAG systems. It first proposes a preference knowledge construction pipeline and incorporates five novel query augmentation strategies to address the issue of preference data scarcity.Based on the preference data, DPA-RAG achieves both external and internal preference alignment. It firstly integrates pairwise, pointwise, and contrastive preference alignment capabilities into the reranker, achieving external preference alignment among RAG components. And it further introduces a pre-aligned stage before traditional supervised fine-tuning (SFT), enabling LLMs to implicitly capture knowledge aligned with their reasoning preferences, thereby achieving internal alignment for LLMs. Experiments are conducted to evaluate the efficiency of the proposed method.

Strong point

S1. The content of the paper is complete and the structure is clear.

S2. A universal framework for aligning diverse knowledge preferences within RAG systems is developed.

S3. Experiments on real-world datasets are conducted to evaluate the proposed algorithms.

Weak point

W1. Although the experiments are comprehensive, the evaluation is mainly focused on the QA dataset. It should be added how DPA-RAG performs on other RAG tasks (such as document summarization or dialogue generation) to verify its generalization, or constrained on the topic, the article only focuses on the QA task.

W2. In the methodology, the concepts such as hierarchically sampling, query augmentation strategies, and LLM self-alignment are already commonly used, lacking innovation in the approach.

W3. In the experiment, the tables are not very readable.

**Questions:**

1. What does the knowledge preference of LLM refer to specifically? When it comes to preferences, it is common to align with human preferences. The knowledge preference mentioned in the text seems to be a vague concept, can you give an example?

2. Does the preference gap refer to retrievers preferring semantic similar documents while LLMs do not prefer or need semantic similar documents? If this is what you said inherent preference gap between the retriever and the LLM-based reader in RAG systems, such a gap seems to be strongly related to the coverage of the retrieved document. How to argue that the existence of this gap is inherent.

3. Why was the effectiveness validated only on the QA benchmark? It can be seen that DPA-RAG outperforms other RAG methods in QA tasks, but the retrieval scope of these tasks is relatively narrow, which seems to make it difficult to demonstrate the universality of the alignment effect for the "preference gap" issue.

**Reviewer Confidence:**

3: The reviewer is confident but not certain that the evaluation is correct

**Scope:**

4: The work is relevant to the Web and to the track, and is of broad interest to the community